# Zero-shot visual reasoning through probabilistic analogical mapping

Taylor Webb ©[1,4] ✉, Shuhao Fu[1,4], Trevor Bihl[2], Keith J. Holyoak[1] & Hongjing Lu ©[1,3] ✉

Human reasoning is grounded in an ability to identify highly abstract commonalities governing superficially dissimilar visual inputs. Recent efforts to develop algorithms with this capacity have largely focused on approaches that require extensive direct training on visual reasoning tasks, and yield limited generalization to problems with novel content. In contrast, a long tradition of research in cognitive science has focused on elucidating the computational principles underlying human analogical reasoning; however, this work has generally relied on manually constructed representations. Here we present visiPAM (visual Probabilistic Analogical Mapping), a model of visual reasoning that synthesizes these two approaches. VisiPAM employs learned representations derived directly from naturalistic visual inputs, coupled with a similarity-based mapping operation derived from cognitive theories of human reasoning. We show that without any direct training, visiPAM outperforms a state-of-the-art deep learning model on an analogical mapping task. In addition, visiPAM closely matches the pattern of human performance on a novel task involving mapping of 3D objects across disparate categories.

If a tree had a knee, where would it be? Human reasoners (without any special training) can provide sensible answers to such questions as early as age four, revealing deep understanding of spatial relations across different object categories[1]. How such abstraction is possible has been the focus of decades of research in cognitive science, leading to several proposed theories of analogical reasoning[2–8]. The basic elements of these theories include structured representations involving bindings between entities and relations, together with some mechanism for mapping the elements in one situation (the source) onto the elements in another (the target), based on the similarity of those elements. However, while this work has succeeded in accounting for a range of empirical signatures of human analogy-making, a significant limitation is that the structured representations at the heart of these theories have typically been hand-designed by the modeler, without providing an account of how such representations might be derived from real-world perceptual inputs[9]. This concern is particularly notable in the case of visual analogies. Unlike linguistic analogies

in which relations are often given as part of the input (via verbs and relational phrases), visual analogies more obviously depend on a mechanism for the eduction of relations[10]: the extraction of relations from non-relational inputs, such as pixels in images.

More recently, a separate line of research inspired by the success of deep learning in computer vision and other areas has aimed to develop neural network algorithms with the ability to solve analogy problems. This work tackles the challenge of solving analogy problems directly from pixel-level inputs; however, it has typically done so by eschewing the structured representations and reasoning operations that characterize cognitive models, focusing instead on end-to-end training directly on visual analogy tasks. These approaches have generally relied on datasets with massive numbers (sometimes more than a million) of analogy problems, and typically yield algorithms that cannot readily generalize to problems with novel content[11–13]. More recent efforts have attempted to address this limitation, proposing algorithms that can learn from fewer examples or display some degree

[1]Department of Psychology, University of California, Los Angeles, USA. [2]Sensors Directorate, Air Force Research Laboratory, Wright Patterson AFB, Ohio, USA. [3]Department of Statistics, University of California, Los Angeles, USA. [4]These authors contributed equally: Taylor Webb, Shuhao Fu.
✉e-mail: taylor.w.webb@gmail.com; hongjing@ucla.edu

of out-of-distribution generalization[14–18]. But this work still maintains the basic paradigm of treating analogy problems as a task on which a reasoner will receive at least some direct training. This is in stark contrast to human reasoners, who can often solve analogy problems zero-shot—that is, with no direct training on those problem types. When visual analogy problem sets are administered to a person, training is typically limited to general task instructions with at most one practice problem. This format is critical to the use of analogy problems to measure fluid intelligence[19,20], the ability to reason about novel problems without prior training or exposure.

Here we propose a synthesis that combines the strengths of these two approaches. Our proposed model, visiPAM (visual Probabilistic Analogical Mapping), combines learned representations derived directly from pixel-level or 3D point-cloud inputs, together with a reasoning mechanism inspired by cognitive models of analogy. Visi-PAM addresses two key challenges that arise in developing such a synthesis. The first issue is how structured representations might be extracted from unstructured perceptual inputs. VisiPAM employs attributed graph representations that explicitly keep track of entities, relations, and their bindings. We propose two methods for deriving such representations, one for 2D images and one for point-cloud inputs representing 3D objects. Second, representations inferred from real perceptual inputs are necessarily noisy and high-dimensional, posing a challenge for any reasoning mechanism that must operate over them. To address this problem, visiPAM employs a recently proposed Probabilistic Analogical Mapping (PAM)[21] method that can efficiently identify patterns of similarity governing noisy, real-world inputs. We evaluated visiPAM on a part-matching task with naturalistic images, where it outperformed a state-of-the-art deep learning model by a large margin (30% relative reduction in error rate)—despite the fact that, unlike the deep learning model, visiPAM received no direct training on the part-matching task. We also performed a human experiment involving visual analogies between 3D objects from disparate categories (e.g., an analogy between an animal and a man-made object), where we found that visiPAM closely matched the pattern of human behavior across conditions. Together, these results provide a proof-of-principle for an approach that weds the representational power of deep learning with the similarity-based reasoning operations that characterize human cognition.

## Results
### Computational framework
Figure 1 shows an overview of our proposed approach. VisiPAM consists of two core components: (1) a vision module that extracts structured visual representations from perceptual inputs corresponding to a source and a target, and (2) a reasoning module that performs probabilistic mapping over those representations. The resulting mappings can then be used to transfer knowledge from the source to the target, to infer the part labels in the target image based on the part labels in the source.

The inputs to the vision module consisted of either 2D images, or point-cloud representations of 3D objects. The vision module uses deep learning components to extract representations in the form of attributed graphs. These graph representations consist of a set of nodes and directed edges, each of which is associated with a set of attributes (i.e., a vector). In the present work, nodes corresponded to object parts, and edges corresponded to spatial relations, though other arrangements are possible (e.g., nodes could correspond to entire objects in a multi-object scene, and edges could encode other visual or semantic relations in addition to spatial relations). For 2D image inputs, we used iBOT[22] to extract node attributes that captured the visual appearance of each object part. iBOT is a state-of-the-art, self-supervised vision transformer that was pretrained on a masked image modeling task. Masked image modeling shows promise as a suitable pretraining objective for capturing general visual appearance, rather than only information relevant to a particular task, such as object classification. For 3D point-cloud inputs, we used a Dynamic Graph Convolutional Neural Network (DGCNN)[23] trained on a part segmentation task in which each point was labeled according to the object part to which it belonged. We hypothesized that this objective would encourage the intermediate layers of the DGCNN (which were used to generate node embeddings) to represent the local spatial structure of object parts. We evaluated visiPAM on analogies involving 3D objects from either the same superordinate category (e.g., both the source and target were man-made objects) or different superordinate categories (e.g., the source was a man-made object, and the target was an animal). Note that the DGCNN was only trained on part segmentation for man-made objects, not for animals. Edge attributes encoded either the 2D or 3D spatial relations between object parts (see "Edge

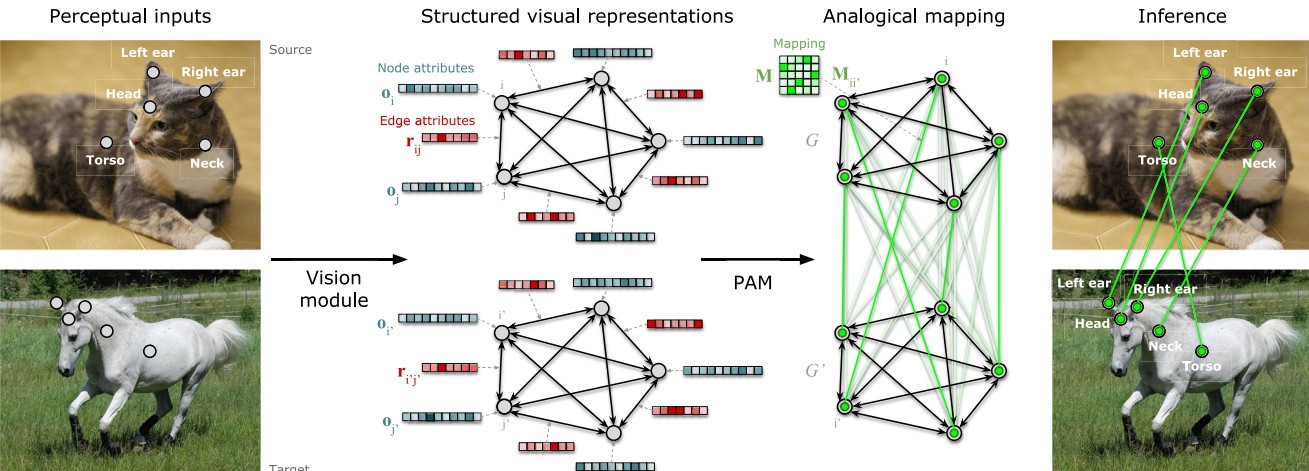

**Fig. 1 | Overview of visiPAM.** VisiPAM contains two core components: a vision module and a reasoning module. The vision module receives visual inputs in the form of either 2D images, or point-cloud representations of 3D objects, and uses deep learning components to extract structured visual representations. These representations take the form of attributed graphs, in which both nodes $\mathbf{o}_{1..N}$ (corresponding to object parts) and edges $\mathbf{r}_{1..N(N-1)}$ (corresponding to spatial relations between parts) are assigned attributes. The reasoning module then uses Probabilistic Analogical Mapping (PAM) to identify a mapping $\mathbf{M}$ from the nodes of the source graph $G$ to the nodes of the target graph $G'$, based on the similarity of mapped nodes and edges. Mappings are probabilistic, but subject to a soft isomorphism constraint (preference for one-to-one correspondences).

embeddings" for more details). We denote the node attributes of each graph as $\mathbf{o}_{i=1..N}$, and the edge attributes as $\mathbf{r}_{1..N(N-1)}$, where $N$ is the number of nodes in the graph. We denote the source graph as $G$ and the target graph as $G'$.

After these representations were extracted by the vision module, we used PAM to identify correspondences between analogous parts in the source and target, based on the pattern of similarity across both parts and their relations. Formally, this corresponds to a graph-matching problem, in which the objective is to identify the mapping $\mathbf{M}$ that maximizes the similarity of mapped nodes and edges. PAM adopts Bayesian inference to compute the optimal mapping:

$$p(\mathbf{M}|G,G') \propto p(G,G'|\mathbf{M})p(\mathbf{M}) \qquad (1)$$

where $\mathbf{M}$ is an $N \times N$ matrix in which each entry $\mathbf{M}_{ii'}$ corresponds to the strength of the mapping between node $i$ in the source and node $i'$ in the target. The likelihood term $p(G,G'|\mathbf{M})$ in Equation (1) is based on the following log-likelihood:

$$\log(p(G,G'|\mathbf{M})) = (1-\alpha)\frac{\sum_i\sum_{j\neq i}\sum_{i'}\sum_{j'\neq i'}\mathbf{M}_{ii'}\mathbf{M}_{jj'}\mathrm{sim}(\mathbf{r}_{ij},\mathbf{r}_{i'j'})}{N(N-1)} + \alpha\frac{\sum_i\sum_{i'}\mathbf{M}_{ii'}\mathrm{sim}(\mathbf{o}_i,\mathbf{o}_{i'})}{N}$$

$$(2)$$

in which $\mathrm{sim}(\mathbf{r}_{ij},\mathbf{r}_{i'j'})$ is the cosine similarity between edge $\mathbf{r}_{ij}$ (the edge between nodes $i$ and $j$ in the source) and edge $\mathbf{r}_{i'j'}$ (the edge between nodes $i'$ and $j'$ in the target), and $\mathrm{sim}(\mathbf{o}_i,\mathbf{o}_{i'})$ is the cosine similarity between node $\mathbf{o}_i$ in the source and node $\mathbf{o}_{i'}$ in the target. Intuitively, the optimal mapping is one that assigns high strength to similar nodes, and to nodes attached by similar edges. The relative influence of node vs. edge similarity is controlled by the parameter $\alpha$. In addition, PAM incorporates a prior $p(\mathbf{M})$ that favors isomorphic (one-to-one) mappings (see detailed equation in section "VisiPAM reasoning module"). In other words, a mapping function with one-to-one correspondences between the source and the target is assigned a higher prior probability. At the algorithm level, given that an exhaustive search over possible mappings was not feasible (we address problems with up to 10 object parts, corresponding to $10! \approx 3.6$ million possible one-to-one mappings), we used a graduated assignment algorithm[24]. In this approach, $\mathbf{M}$ is initialized as a uniform mapping, in which each source node is equally likely to map to any target node, and this mapping is then iteratively updated based on the likelihood in Eq. (2). This iterative procedure is also guided by the prior $p(\mathbf{M})$ to encourage convergence to an isomorphic (one-to-one) mapping. It should be noted that this approach is fully differentiable, and therefore the entire model, including the vision module, could in principle be learned end-to-end based on the mapping task. However, since our goal in the present work was to account for the human capacity for zero-shot analogical reasoning, we instead use visual representations provided from pretrained deep learning models, such that no direct training on the mapping task is required. More details about PAM can be found in Section 4.3.

### Analogical mapping with 2D images

We evaluated visiPAM on a part-matching task developed by Choi et al.[25]. In this task two images are presented, each together with a set of coordinates corresponding to the locations of various object parts (as in Fig. 1). The task is to label the object parts in the target image, based on a comparison with the labeled object parts in the source image. Importantly, the test set for this dataset consists only of object categories not present in the training set, so that it is not possible to solve the task by learning to classify the object parts in the target image directly. Choi et al. also developed the Structured Set Matching Network (SSMN, see Section 4.4), which they found outperformed a range of other methods after training directly on the part-matching task with a set of separate object categories. Here we go beyond this benchmark

**Table 1 | Analogical mapping with 2D images**

| | Within-category | | Between-category |
| --- | --- | --- | --- |
| | **Animals** | **Vehicles** | **Animals** |
| **VisiPAM** | **63.2% (59.1%)** | **69.5% (58.8%)** | **67.9% (59.9%)** |
| VisiPAM (nodes only) | 55.5% (50.6%) | 61.6% (48.1%) | 62.3% (52.9%) |
| VisiPAM (edges only) | 47.8% (42%) | 63.7% (50.9%) | 51.5% (39.4%) |
| SSMN | 46.6% (40.7%) | | |
| Random | 10% | 26% | 20% |

Mapping accuracy on part-matching task proposed by Choi et al.[25]. Original task in ref. 25 involved the evaluation on within-category animal comparisons after training with 37,330 mapping problems. VisiPAM significantly outperformed SSMN, the previous state-of-the-art, despite having no direct training on mapping. VisiPAM also performed well on new problems involving within-category vehicle comparisons, and between-category animal comparisons (e.g., mapping from cat to horse). VisiPAM performed best when mapping was based on both node and edge similarity (as indicated by bold text). 'Random' denotes chance performance (determined by average number of part comparisons). Values in parentheses reflect chance-normalized performance (percentage of the range between chance performance and 100% accuracy). VisiPAM's performance is roughly comparable across all conditions once chance performance is taken into account.

by removing any direct training on the part-matching task, relying instead on visiPAM's similarity-based reasoning mechanism to align the object parts between source and target.

Table 1 shows the results of this evaluation. Accuracy was computed based on the proportion of parts that were mapped correctly across all problems (i.e., the model could receive partial credit for mapping some but not all of the parts correctly in a given image pair). As originally proposed, the part-matching task involved within-category comparisons of animals (comparing either two images of cats, or two images of horses). In that setting, visiPAM significantly outperformed SSMN, resulting in a 30% reduction relative to SSMN's error rate, despite the fact that SSMN, but not visiPAM, received direct training (37,330 problems) on this task. We also developed extensions of this task involving other object categories. These included within-category mapping of vehicles (involving either two images of planes, or two images of cars), and between-category mappings of animals (mapping from an image of a cat to an image of a horse). We found that visiPAM performed comparably well across all three of these conditions (within- and between-category animal mapping, within-category vehicle mapping).

A key element of our proposed model is that it performs mapping based on the similarity of both object parts and their relations. To determine the importance of this design decision, we performed ablations on either node or edge similarity components, by setting $\alpha$ to either 0 or 1. We found that both of these ablations significantly impaired the performance of visiPAM. This pattern aligns with findings from studies of human analogy-making, which show that human reasoners are typically sensitive to similarity of both entities and relations[26].

We also performed ablations that targeted different aspects of visiPAM's edge embeddings. Specifically, visiPAM's edge embeddings contain information based on both angular distance and relative location (vector difference). We found that both of these sources of information contributed to visiPAM's performance (Supplementary Table S2). Furthermore, we found that augmenting visiPAM's edge embeddings with topological information (i.e., whether two parts are connected) can lead to further improvements in mapping performance (Supplementary Table S3). Thus, the specific types of spatial relations employed by the model play an important role in its performance, and the addition of new sources of relational information will likely lead to continued improvement.

Figure 2 shows some examples of the mappings produced by visiPAM. These include some impressive successes, including mapping

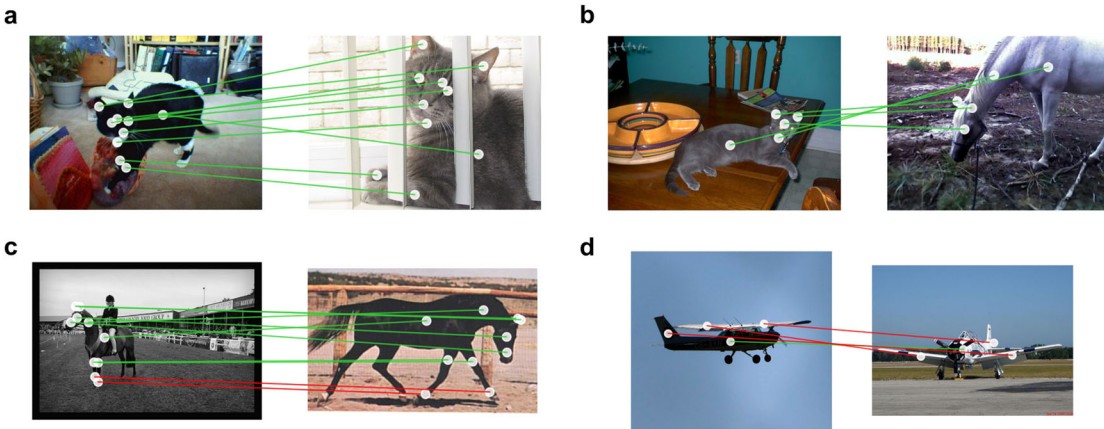

**Fig. 2 | VisiPAM part mappings with 2D images.** Examples of part mappings identified by visiPAM for 2D image pairs. **a** Successful within-category animal mapping involving 10 separate parts. Note the significant variation in visual appearance between source and target. **b** Successful between-category animal mapping. **c** Within-category animal mapping illustrating a common error in which the left and right feet are mismapped. **d** Within-category vehicle mapping. Mapping between images of planes was especially difficult, likely due to significant variation in 3D pose.

of a large number of object parts across dramatic differences in background, lighting, pose, and visual appearance (Fig. 2a), as well as between objects of different categories (Fig. 2b). Notably, we also found that visiPAM's performance was unaffected when the target image was horizontally reflected relative to the source (Supplementary Section S2.1 and Table S1), suggesting that the model is robust to low-level image manipulations.

Our analyses also illustrated some of the model's limitations. Figure 2c shows an example of an error pattern that we commonly observed, in which visiPAM confused corresponding left and right parts (in this case the left and right feet in a comparison of two horses). A more comprehensive analysis is shown in Supplementary Figs. S2–S4, where it can be seen that this kind of confusion of corresponding lateralized parts accounts for a large percentage of visiPAM's errors. We also found that visiPAM particularly struggled with mapping images of planes (achieving the model's lowest within-category mapping accuracy of 42.5%), as shown in Fig. 2d. This is likely due to the limitation of the 2D spatial relations used in this version of the model, which are particularly ill-suited to objects such as planes that can appear in a wide variety of poses and viewpoints. Thus, while visiPAM shows an impressive ability to perform mappings between complex, real-world images using only these 2D spatial relations, we suspect that accurate 3D spatial knowledge is likely necessary to solve some challenging visual analogy problems at a human level. To address this concern, we next sought to evaluate the performance of visiPAM in the context of 3D visual inputs.

**Analogical mapping between 3D objects**

We evaluated visiPAM on analogies involving point-cloud representations of 3D objects, and compared visiPAM's performance with the results of a human experiment.[27] In that experiment, we presented human participants with analogy problems consisting either of images from the same superordinate category (e.g., an analogy between a dog and a horse), or two different superordinate categories (e.g., an analogy between a chair and a horse). Human behavioral data was especially important as a benchmark in the latter case, since there is not necessarily a well-defined, correct mapping between the parts of the two objects.

Figure 3a shows the analogy task used in the human experiment, which we adapted from a classic task employed by Gentner[1]. On each trial, participants received a source image and a target image. The source image had two colored markers placed on the object, while the target image had two markers that appeared in the upper right corner. Participants were instructed to 'move the marker on the top right

corner in the target image to the corresponding location that maps to the same-color marker in the source image.' Figure 3b shows two representative examples of human responses. Marker placements from different participants are shown as a heatmap on the target image. When presented with source and target images from the same superordinate categories, there was generally strong agreement between participants. Responses were significantly more variable when participants were presented with source and target images from different superordinate categories (distance from mean marker placement of 32.06 pixels in the different-superordinate-category condition vs. 8.08 pixels in the same-superordinate-category condition; 2 (target category, animal vs. man-made object) X 2 (category consistency, different- vs. same-superordinate-category) repeated-measures ANOVA, main effect of category consistency $F_{1,40} = 625.37$, $p < 0.0001$). We also found a significant interaction, such that the effect of category consistency was larger for problems where the target category was an animal vs. a man-made object ($F_{1,40} = 19.29$, $p < 0.0001$). There was no significant main effect of target category. For problems involving mapping between different superordinate categories, responses in some trials were bimodal, with some participants preferring one mapping (e.g., from the back of the chair to the head of the horse), and other participants preferring another mapping (e.g., from the back of the chair to the back of the horse), though the same qualitative effects were still present even after accounting for these separate clusters (main effect of category consistency, $F_{1,40} = 151.53$, $p < 0.0001$; interaction effect, $F_{1,40} = 11.91$, $p < 0.005$; see section "Analysis" for more details on clustering analysis).

Figure 4 shows some examples of mappings produced by visiPAM for point-cloud representations of the 3D objects used in the human experiment. To apply visiPAM to these problems, we obtained embeddings for each point using the DGCNN, then clustered these points, and assigned each cluster to a node, where the attributes of that node were defined based on the average embedding for all points in the cluster. Edge attributes were defined based on the 3D spatial relations between cluster centers. Figure 4 shows examples of successful mappings for all four problem types, though visiPAM also sometimes made errors (see Supplementary Section S2.2 for detailed error analysis).

To systematically compare the model's behavior with human responses, we applied the model to all image pairs used in the experiment, and measured the distance from the marker location predicted by visiPAM to the mean locations of human placements for each pair. Figure 5 shows the results of this comparison. We found that visiPAM reproduced the qualitative pattern displayed by human

**Fig. 3 | Experiment measuring human performance for mapping between 3D objects. a** Sample stimuli used in human experiment. Participants were instructed to move markers in target image to locations corresponding to same-color markers in source image. Left panel: example trials with source and target images from the same superordinate object category. Right panel: example trials with images from different superordinate categories. **b** Example heatmaps of human marker placements on target images for two comparisons. The intensity of the color indicates the proportion of participants who placed the marker in that location. Source images have been reduced in size for the purpose of illustration. Left panel: marker placements were highly consistent across subjects when source and target came from same superordinate category. Right panel: marker placement showed more variation across participants when source and target came from different superordinate categories.

mappings across conditions. As with the human participants, visiPAM's mappings showed greater deviation from the mean human placement when mapping between objects from different vs. same superordinate categories. The size of this difference was also larger when the target category was an animal vs. man-made object, capturing the significant interaction effect observed in the human data (see Supplementary Fig. S6 for an explanation of this interaction). Overall, marker locations of analogous parts predicted by visiPAM were an average of 25 pixels from mean locations of human placements, close to the average human distance to mean locations (20 pixels). Relative to the object sizes (average height of 213 pixels and width of 135 pixels), the model and human distances were very similar. The same qualitative results were present when taking the bimodal nature of human responses into account (i.e., when measuring distance to the mean of the closest human cluster for trials with a bimodal response distribution), with both visiPAM and human participants showing lower variance when behavior was quantified in this manner (Supplementary Fig. S8).

We also calculated the item-level correlation across all analogy problems between average human distances from mean placement locations and distances of the model predictions from the same mean locations. VisiPAM reliably predicted human responses at the item level ($r = 0.70$). In addition, as with the results for analogies between 2D images, we found that visiPAM performed best when mapping involved both node and edge similarity (Supplementary Table S4). The ability of visiPAM to predict human responses at the item level was impaired both when focusing exclusively on node similarity ($r = 0.61$ for $\alpha = 1$), and when focusing exclusively on edge similarity ($r = 0.60$ for $\alpha = 0$).

## Discussion

We have presented a computational model that performs zero-shot analogical mapping based on rich, high-dimensional visual inputs. To accomplish this, visiPAM integrates representations derived using general-purpose algorithms for representation learning together with a similarity-based reasoning mechanism derived from theories of human cognition. Our experiments with analogical mapping of object parts in 2D images showed that visiPAM outperformed a state-of-the-art deep learning model, despite receiving no direct training on the analogy task. Our experiment with mapping of 3D object parts showed that, when armed with rich 3D visual representations, visiPAM matched the pattern of human mappings across conditions.

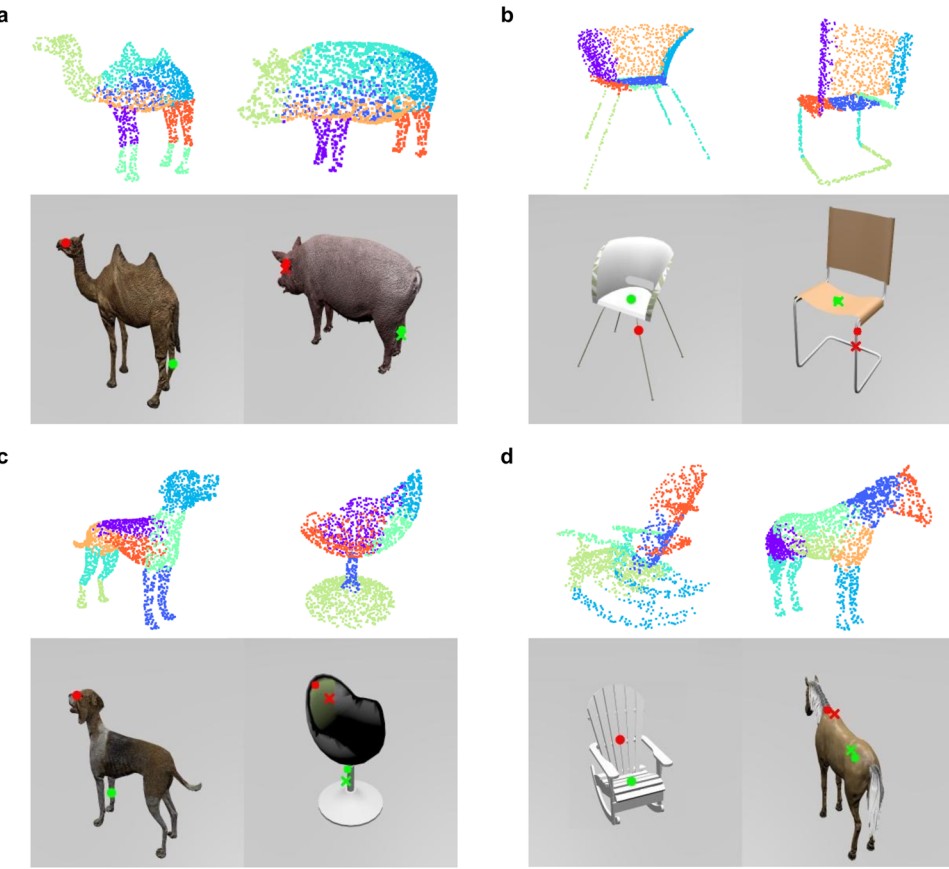

**Fig. 4 | VisiPAM part mappings between 3D objects.** Examples of part mappings for 3D objects represented as point-clouds. Colored regions of point-clouds (upper panels in each figure) indicate the mappings identified by visiPAM for each cluster in the source and target. Images (lower panels in each figure) depict the source marker locations (red and green circles in the lower left panels of each figure), the target marker locations identified by visiPAM (red and green circles in the lower right panels of each figure) and the average of the target marker locations identified by human participants (red and green crosses). **a, b** Examples of problems in which the source and target are from the same superordinate category. **c, d** Examples of problems in which the source and target are from different superordinate categories.

These results build on other recent work that has employed similarity-based reasoning over representations derived using deep learning[21,28,29], enabling analogy to emerge zero-shot. Most notably, visiPAM builds on a previous model of analogical mapping, PAM[21], by combining it with embeddings from state-of-the-art computer vision models[22,23], thus enabling its application to real-world visual inputs, even when those inputs are from distinctively different categories (i.e., far analogy). VisiPAM also has commonalities with some recent deep learning models. These include models of visual analogy that are applied to real-world visual inputs[25,30–32], and models that incorporate graph-based representations similar to visiPAM's structured visual representations[33–35]. Unlike visiPAM, these deep learning models all involve end-to-end training on a large number of examples for a specific task, with limited generalization to new analogy tasks. Among deep learning models, visiPAM is most closely related to models that incorporate a relational bottleneck, notably involving a central role for similarity-based mechanisms[16–18]. These models achieve better out-of-distribution generalization in relational tasks, and require significantly fewer training examples than standard deep learning models, but unlike visiPAM they are still centered around end-to-end training on specific tasks, and therefore are not capable of zero-shot reasoning on completely novel tasks.

A few recent studies have applied a more traditional cognitive model of analogical mapping, the Structure-Mapping Engine (SME), to visual analogy problems involving simple geometric forms[36,37]. This work represents an impressive effort, showing that SME is able to solve the majority of problems in the Raven's Standard Progressive Matrices[38], a commonly used measure of fluid intelligence. However, a significant limitation of this approach lies in its traditional symbolic representations, which are derived from image-based inputs using hand-designed algorithms. This characteristic arguably limits the ability of such an approach to address more complex, real-world inputs such as natural images. Other so-called neurosymbolic methods have attempted, with some success, to extract symbolic representations directly from pixel-level inputs, which can then be passed to a task-specific symbolic algorithm[39]. However, these approaches may not be able to capture the richness of real-world perceptual inputs, or the wide variety of tasks that human reasoners can perform over those inputs. Here, by contrast, we have specifically sought to employ representations that preserve as much of this perceptual richness as possible (by representing elements as vectors), while also incorporating the structured nature of symbolic representations (by maintaining explicit bindings between entities and relations). This was made possible by using a probabilistic reasoning algorithm capable of accommodating such high-dimensional, uncertain inputs.

Human analogical reasoning is thought to be driven by similarity both at the relational level and at the basic object level[26]. This latter influence has often been framed as a deficiency in human reasoning—an inability to achieve the pure abstraction that would presumably be enabled by focusing exclusively on relations. An important finding from the present study is that visiPAM performed best when constrained both by edge (relational level) and node (object level) similarity. This result suggests that the influence of object-level similarity in human analogical reasoning may be driven in part by the fact that it is a useful constraint in the context of complex, real-world analogy-making[40].

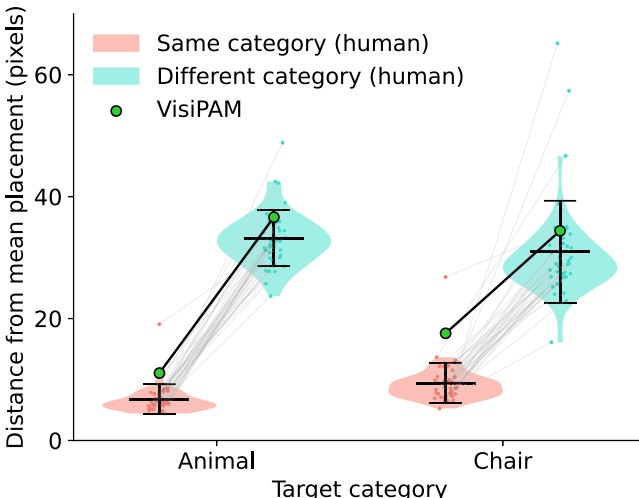

**Fig. 5 | Comparison of visiPAM with human behavior.** Violin plot comparing human placements and visiPAM predictions. Strip plots (small colored dots) indicate average distance of marker locations from the overall human mean, one point for each participant ($N = 41$). Thin gray lines show the within-participant differences for the same- vs. different-superordinate-category conditions. Violin plots exclude data points > 2.5 standard deviations away from mean, though these individual data points are included in strip plots. Black horizontal lines indicate mean human distances in each condition, and error bars indicate standard deviations. Large green dots indicate visiPAM predictions (i.e., the distance between model predicted location and human mean location). Both human and visiPAM mappings were more variable when mapping objects from different superordinate categories.

One limitation of the present work is the lack of a method for deriving 3D representations (such as point-clouds) from the 2D inputs that human reasoners typically receive. We found the incorporation of topological information (e.g., whether two parts are connected) significantly improved visiPAM's mapping performance, but it would be desirable to derive this information directly from 2D inputs. Some recent advances may make it possible to take a step in that direction[41], but visiPAM will likely benefit from the rapid pace of innovation in representation learning algorithms. We are particularly excited about the continued development of general-purpose, self-supervised algorithms (such as the iBOT algorithm that we employed[22]), which we hope will provide increasingly rich representations over which reasoning algorithms such as PAM can operate. Similarly, it will be useful in future work to incorporate non-visual semantic and functional information into visiPAM's representations. We found that this kind of information was a significant contributing factor to human mapping judgments. Potential sources of such knowledge include word embeddings[42], multimodal embeddings[43], and representations of relations between verbal concepts[28].

Finally, an important next step will be to expand the capabilities of visiPAM to incorporate other basic cognitive elements of analogy. For example, human analogical reasoning often involves an interaction between bottom-up and top-down processes, in which the reasoning process can sometimes guide the reinterpretation of perceptual inputs[5,9,44]. VisiPAM in its current formulation is purely bottom-up—visual representations are constructed by the vision module, and then passed to PAM to perform mapping—but a promising avenue for future work will be to incorporate top-down feedback to the vision module. One possibility for doing so is to incorporate additional parameters (e.g., parameters for visual feature attention) to visiPAM's optimization procedure, such that the visual representations are modulated based on the downstream graph-matching objective. In addition, the present work focused on the identification of analogies between pairs of images, but human reasoners are also capable of inducing more general schematic representations that capture abstract similarity across

multiple examples[45]. Future work should investigate how visiPAM's reasoning mechanisms can be extended to capture this capacity for schema induction. There are many exciting prospects for the continued synergy between rich visual representations and human-like reasoning mechanisms.

## Methods
### Datasets
**Analogical mapping with 2D images.** To evaluate visiPAM on analogical mapping with 2D images, we used the Pascal Part Matching (PPM) dataset proposed by Choi et al.[25]. This dataset was developed based on the Pascal Part dataset[46], a dataset containing images of common object categories, together with segmentation masks for object parts. PPM consists of images from 6 categories (cow, dog, person, sheep, cat, and horse). Each problem consists of a pair of images from the same category, together with coordinates corresponding to object parts, as well as the labels associated with those parts. The task is to label the parts in the target image based on a comparison with the labeled parts in the source image. Choi et al. trained various models on 37,330 problems consisting of images from four object categories (cow, dog, person, and sheep), and tested those models on 9060 problems consisting of images from two distinct object categories (cat and horse). In the present work, we did not train visiPAM on part matching at all. Therefore, we only used the problems in the test set. For these problems, both the source and target image each contained 10 object parts. There were 5860 problems involving cat images, and 3200 problems involving horse images. Accuracy was computed based on the proportion of parts that were mapped correctly across all problems (i.e., the model could receive partial credit for mapping some but not all of the parts correctly in a given image pair). Despite the class imbalance between problems involving horses and cats, we computed performance on this dataset as an average over all problems, to be consistent with the method used by Choi et al., and therefore ensure a fair comparison with their SSMN method. The PPM dataset can be downloaded from: https://allenai.github.io/one-shot-part-labeling/ppm/.

We also created two variants on the PPM dataset. First, we created problems involving between-category comparisons of animals. To do this, we selected the subset of object parts shared by both the cat and horse images (head, neck, torso, left and right ear), and created 3200 problems in which the source was an image of a cat, and the target was an image of a horse. Second, we created new problems involving within-category comparisons of vehicles. To do so, we used images of cars and planes from the original Pascal Part dataset. We generated markers corresponding to the centroid of the masks for each object part. Then, we identified within-category image pairs for which there were at least two common object parts. Some parts had to be excluded due to a lack of systematic correspondences between images (for instance, wheels are sometimes labeled, but there does not appear to be a consistent numbering scheme that might allow identification of corresponding wheels in two images). This procedure identified 406 mapping problems involving images of cars, and 6860 problems involving images of planes, with an average of 3.85 parts per problem. Due to this significant class imbalance, we first computed the average performance for cars and the average performance for planes, and then computed an average of these two values.

**Analogical mapping with 3D objects.** 3D object stimuli were selected from two publicly available datasets used in computer vision: ShapeNetPart[47] and a 3D animal dataset (Animal Pack Ultra 2) from Unreal Engine Marketplace, which are shown in Supplementary Fig. S1. Eight chairs were selected from the ShapeNetPart dataset (each chair with a different shape), and eight animals from the Animal Pack dataset: horse, buffalo, Cane Corso, domestic pig, Celtic wolfhound, African elephant, Hellenic hound, and camel. For simulations with

visiPAM, the vision module (described in Section "Node embeddings for 3D objects") received point-clouds sampled directly from these 3D object models, using the CloudCompare software (v2.12). For the human experiment, we used the Blender software (v2.79) to render 2D images from the 3D models. The stimulus images were generated using a constant lighting condition, with a gray background. Multiple camera positions were sampled for each object, with 30° separation between camera angles for depth rotation. Two undergraduate research assistants manually annotated the keypoints (i.e., center locations) of pre-defined parts on the 3D objects, using the Unreal Engine software. Chair parts included seat, back, and chair legs, while animal parts included spine of torso, head, and legs.

We generated 192 pairs of images. Each image pair included a source image (either a chair or an animal), which was annotated with two markers on two different parts of the object. To generate marker locations for source images, we first rendered the images using the corresponding 3D object models, and then calculated marker locations on the rendered 2D images using a perspective projection for the predefined camera position. In one condition, the source and target images were from the same superordinate object category (e.g., two images of animals). In another condition, the two images were from different superordinate object categories (e.g., a chair image with an animal image). The two objects in an image pair were shown in the same orientation.

## VisiPAM vision module

**Node embeddings for 2D images.** We used iBOT[22] to extract node attributes for object parts from 2D images. iBOT is a self-supervised vision model, trained on a masked image modeling task. Masked image modeling (MIM) is similar to the masked language model pre-training approach that has become popular in natural language processing[48], in which a neural network model (typically a transformer[49]) is trained to fill in masked tokens in an input sequence. Similarly, MIM is the task of filling in masked patches in an input image. Predicting masked patches directly in image space is significantly more challenging than masked language modeling, presumably due to the amount of fine detail present in pixel-level representations relative to linguistic tokens. To address this, iBOT uses an online tokenization scheme in which the MIM objective is applied in a learned token space, rather than directly in pixel space. iBOT was trained on the ImageNet dataset[50], and is currently the state-of-the-art approach for unsupervised image classification on that dataset (the ability to linearly decode image classes given embeddings learned without supervision).

We used the pre-trained version of iBOT that employs a vision transformer (ViT) architecture[51]. We specifically used the version that employs the largest variant of this architecture, ViT-L/16, and was trained on ImageNet-1K using a random (as opposed to block-wise) masking scheme. We downloaded the pre-trained parameters for this model from: https://github.com/bytedance/ibot. To apply iBOT to the images from the PPM dataset, we resized the images to 224 × 224, and split them into 16 × 16 patches. We then passed the images through the pre-trained iBOT model, and obtained the 1024-dimensional embeddings for these patches in the final layer of the transformer backbone (not the projection head). We used bilinear interpolation to obtain embeddings corresponding to the specific coordinate locations for each object part.

These simulations were carried out in Python v3.9.12 using PyTorch v2[52], NumPy v1.19.2[53], SciPy v1.6.2[54], and code from the iBOT GitHub repository (linked to above).

**Node embeddings for 3D objects.** We used a Dynamic Graph Convolutional Neural Network (DGCNN)[23] to extract node attributes for object parts from 3D point-clouds. The core component of DGCNN is the EdgeConv operation: for each point, this operation first computes edge embeddings for that point and each of its $K$ nearest neighbors

(using a shared Multilayer Perceptron (MLP) that takes pairs of points as input), then aggregates these edge embeddings. Nearest neighbors are recomputed after each EdgeConv operation based on the resulting embeddings (thus making the graph dynamic). The point-cloud first passes through three layers of EdgeConv operations. The features created by each EdgeConv layer are max-pooled globally to form a vector, and concatenated with each other to combine geometric properties. These features are then passed to four additional MLP layers to produce a segmentation prediction for each 3D point.

We used a pre-trained DGCNN, with code that can be downloaded from (the 'part segmentation' model): https://github.com/antao97/dgcnn.pytorch and parameters that can be downloaded from: https://github.com/antao97/dgcnn.pytorch/blob/master/pretrained/model.partseg.t7.

The DGCNN was trained on a supervised part segmentation task[47] using 16 types of 3D objects drawn from the ShapeNetPart dataset, which contains about 17,000 models of 3D objects from 16 man-made object categories, including cars, airplanes, and chairs. Each 3D object is annotated with 2-6 parts. After training with a part segmentation task, DGCNN is able to extract local geometric properties from nearby 3D points and encode these as embedding features. Hence, DGCNN transforms the three-dimensional input (x, y, z coordinates) of each 3D point of the object into a 64-dimensional embedding vector in the third EdgeConv layer. These embeddings capture critical local geometric properties of 3D shapes, and thus represent informative visual features associated with object parts.

In all simulations with visiPAM, the DGCNN was trained on the ShapeNetPart dataset only. Critically, the DGCNN was only trained on man-made objects in the ShapeNetPart dataset and was never trained with 3D animals. About 2000 points were used to represent each 3D object. To reduce the computation cost in the reasoning module, a clustering algorithm (KMeans++ algorithm[55]) was applied to point embeddings to group the points into eight clusters. These clusters tended to correspond to a semantically meaningful part of the object. Node attributes were defined based on the average embeddings for each cluster.

The DGCNN was implemented in Python v3.9.12 using PyTorch v2[52], NumPy v1.19.2[53], and scikit-learn v1.0.2[56].

**Edge embeddings.** Edge embeddings were computed based on the spatial relations between object parts. For the analogy problems with 2D images, these spatial relations were computed using the 2D part coordinates. We defined two types of spatial relations, one based on angular distance:

$$\mathbf{r}_{\theta_{ij}} = [\cos\theta(\mathbf{c}_i - \mathbf{c}_j, \mathbf{c}_i - \mathbf{c}_0), \cos\theta(\mathbf{c}_i - \mathbf{c}_0, \mathbf{c}_j - \mathbf{c}_0), \cos\theta(\mathbf{c}_i - \mathbf{c}_j, \mathbf{c}_j - \mathbf{c}_0)] \quad (3)$$

and one based on vector difference:

$$\mathbf{r}_{\delta_{ij}} = \frac{[\mathbf{c}_j - \mathbf{c}_i, \mathbf{c}_i - \mathbf{c}_0, \mathbf{c}_j - \mathbf{c}_0]}{\max(\mathbf{c}_{1..N}) - \min(\mathbf{c}_{1..N})} \quad (4)$$

where $\mathbf{c}_i$ and $\mathbf{c}_j$ are the coordinates of nodes $i$ and $j$, $\mathbf{c}_0$ are the coordinates for the centroid of the entire object (defined in terms of the segmentation masks for the object parts), $\cos\theta$ is cosine distance, [,] indicates concatenation, and $\max(\mathbf{c}_{1..N}) - \min(\mathbf{c}_{1..N})$ is the elementwise range of the coordinates for all nodes. For each pair of nodes $i$ and $j$, edge attributes were formed by concatenating $\mathbf{r}_{\theta_{ij}}$ and $\mathbf{r}_{\delta_{ij}}$.

For the analogy problems with 3D images, the coordinates of each cluster were computed based on the average of the 3D coordinates for each point in that cluster, and the object centroid was computed based on the average of the 3D coordinates for all points in the object. Edge attributes were computed using the 3D equivalent of $\mathbf{r}_{\theta_{ij}}$ and $\mathbf{r}_{\delta_{ij}}$.

## VisiPAM reasoning module: probabilistic analogical mapping (PAM)

As described in Section "Computational framework", PAM[21] adopts Bayesian inference to compute the optimal mapping between two attributed graphs $G$ and $G'$, as formulated in Eqs. (1) and (2). The log-likelihood in Eq. (2) incorporates a parameter $\alpha$ that controls the relative influence of node vs. edge similarity, where the mapping is entirely driven by node similarity when $\alpha = 1$, and is entirely driven by edge similarity when $\alpha = 0$. Note that the edge similarity and node similarity terms are normalized by the number of edges ($N(N-1)$) and the number of nodes ($N$), respectively, so that they are on the same scale before being multiplied by $\alpha$. We used a value of $\alpha = 0.9$ for our primary experiments with 2D image mapping and 3D object mapping. We also performed ablation experiments with values of $\alpha = 1$ and $\alpha = 0$.

Importantly, PAM also employs a prior that favors isomorphic (one-to-one) mappings:

$$p(\mathbf{M}) = e^{\frac{1}{\beta}\sum_i \sum_{i'} \mathbf{M}_{ii'} \log \mathbf{M}_{ii'}} \tag{5}$$

where $\beta$ is a parameter that controls the strength of the preference for isomorphism (higher values of $\beta$ correspond to a stronger preference for isomorphism).

To implement the inference in Eq. (1), we used a graduated assignment algorithm[24] that minimizes the following energy function (equivalent to maximizing the posterior in Equation (1), subject to the prior in Eq. (5)):

$$E(\mathbf{M}) = -(1-\alpha)\frac{\sum_i \sum_{j\neq i}\sum_{i'}\sum_{j'\neq i'} \mathbf{M}_{ii'}\mathbf{M}_{jj'}\,\mathrm{sim}(\mathbf{r}_{ij},\mathbf{r}_{i'j'})}{N(N-1)} \\ -\alpha\frac{\sum_i\sum_{i'}\mathbf{M}_{ii'}\,\mathrm{sim}(\mathbf{o}_i,\mathbf{o}_{i'})}{N} - \frac{1}{\beta}\sum_i\sum_{i'}\mathbf{M}_{ii'}\log\mathbf{M}_{ii'} \tag{6}$$

The algorithm starts with a low value of $\beta$, and gradually increases it so as to gradually approximate the one-to-one constraint. We used 500 iterations for all experiments with 2D images, and 200 iterations for all experiments with 3D point-clouds. For all experiments, we used an initial $\beta$ value of $\beta_0 = 0.1$, and an initial mapping matrix $\mathbf{M}_0$ corresponding to a uniform mapping (such that the strength of each mapping was set to $1/N$). Algorithm 1 provides pseudocode for the mapping algorithm. Note that, in the algorithm, the edge similarity term is normalized by the number of edges per node ($2(N-1)$), rather than the total number of edges, since the algorithm computes the compatibility of each potential node-to-node mapping separately (rather than summing across all potential mappings as in Eq. (6)). Before performing mapping, we applied an iterative bistochastic normalization procedure (with 10 iterations) to both the node similarity and edge similarity matrices[57].

We used the final mappings produced by PAM in the following ways. For each target node $i'$, we identified the source node $i$ with the strongest mapping strength. For analogy problems with 2D images, we used this mapping to transfer labels from the source nodes to their corresponding target nodes. For analogy problems with 3D point-clouds, we used the mapping result to generate marker locations in the target object corresponding to the matched markers in the source object. First, we assigned each source marker to one of the clusters (i.e., object parts) in the source object, based on the distance of the 3D coordinates for the marker to the centers of each cluster. Then we computed the location of the target marker by selecting from amongst the points (in the point-cloud) located within the corresponding target cluster (as identified by PAM). For each point, we computed three distances: (1) $d_{local}$, the spatial distance of the point to the center of the matched cluster, (2) $d_{global}$, the spatial distance of the point to the center of the object calculated as the mean of all cluster centers, and (3) $d_{feat}$, the feature distance (Euclidean distance in feature space) to the center of the matched cluster. We computed these distances for the source marker, and for all points in the matched cluster in the target object, and selected the target point that minimized the following metric:

$$d_{\delta_{ii'}} = \frac{1}{3}(|d_{local_i} - d_{local_{i'}}| + |d_{global_i} - d_{global_{i'}}| + |d_{feat_i} - d_{feat_{i'}}|) \tag{7}$$

where $d_{local_i}$, $d_{global_i}$, and $d_{feat_i}$ are the distances for the source marker, and $d_{local_{i'}}$, $d_{global_{i'}}$, and $d_{feat_{i'}}$ are the distances for each target point $i'$. We then used the camera angle for the images presented to human participants to project the 3D coordinates for the selected target point onto the 2D images for comparison with human marker placements.

## Structured set matching network

We compared visiPAM's performance to the results reported by Choi et al.[25] for their Structured Set Matching Network (SSMN). The SSMN has some interesting commonalities, as well as some important differences, with visiPAM. Briefly, the SSMN operates by assigning a score to a specified mapping between source and target parts. This score is based on a combination of: (1) the similarity of the learned embeddings for the mapped parts, (2) a score assigned (by a learned neural network) to spatial relation vectors for mapped parts, (3) a score assigned (again by a learned neural network) to appearance relations for mapped parts, and (4) a hard isomorphism constraint that guarantees only one-to-one mappings are considered. VisiPAM differs from SSMN in the following ways. First, rather than learning representations end-to-end in the service of the part-mapping task, as is done in SSMN, visiPAM employs representations learned in the context of more general-

---

**Algorithm 1: Graduated assignment algorithm used in PAM.**

$\beta \leftarrow \beta_0$ ; // Initialize $\beta$
$\mathbf{M} \leftarrow \mathbf{M}_0$ ; // Initialize M
**for** *iterations* **do**
 $\forall i \in G, \forall i' \in G'$
 $\mathbf{Q}_{ii'} \leftarrow (1-\alpha)\frac{\sum_{j\neq i}\sum_{j'\neq i'}\mathbf{M}_{ii'}\mathbf{M}_{jj'}\,\mathrm{sim}(\mathbf{r}_{ij},\mathbf{r}_{i'j'})}{2(N-1)} + \alpha\mathbf{M}_{ii'}\,\mathrm{sim}(\mathbf{o}_i,\mathbf{o}_{i'})$ ; // Compute compatibility matrix Q
 $\forall i \in G, \forall i' \in G'$
 $\mathbf{M}_{ii'} \leftarrow e^{\beta\mathbf{Q}_{ii'}}$ ; // Update soft assignments
 $\forall i \in G, \forall i' \in G'$
 $\mathbf{M}_{ii'} \leftarrow \frac{\mathbf{M}_{ii'}}{\sum_j \mathbf{M}_{ji'}}$ ; // Normalize M across rows
 $\forall i \in G, \forall i' \in G'$
 $\mathbf{M}_{ii'} \leftarrow \frac{\mathbf{M}_{ii'}}{\sum_{j'} \mathbf{M}_{ij'}}$ ; // Normalize M across columns
 $\beta \leftarrow \beta + \frac{\beta_0}{10}$ ; // Update $\beta$
**end**

---

purpose objectives, either self-supervised learning in the case of our experiments with 2D images, or part segmentation in our experiment with 3D objects. Second, whereas SSMN scores relation similarity between source and target using a learned neural network, visiPAM explicitly computes the similarity of mapped relations. Together, these two features allow visiPAM to perform mapping without any direct training, whereas SSMN relies on learned components that have the opportunity to overfit to the specific examples observed during training. Finally, SSMN is designed to assign a score to a prespecified, one-to-one mapping, necessitating a search over deterministic mappings at inference time. VisiPAM, by contrast, employs a continuous relaxation of this search problem that allows it to much more efficiently converge on a soft, but approximately isomorphic, mapping.

### Human experiment

**Participants.** Fifty-nine participants (mean age = 20.55 years; 51 female) were recruited from the Psychology Department subject pool at the University of California, Los Angeles. All participants provided informed consent, and were compensated with course credit. The study was approved by the Institutional Review Board.

Five out of the 59 participants were removed from analysis either because they indicated they were not serious, or because they moved fewer than 30% of the markers in the entire experiment. Thirteen additional participants were removed because they did not move any of the markers in at least one of the conditions. Thus, data from a total of 41 participants were included in the analyses.

**Procedure.** We collected behavioral data for the 3D object mapping task using an online experiment coded in JavaScript. Each participant performed mapping for all 192 image pairs that were used to evaluate visiPAM. The experiment used a 2 (target category, animal vs. man-made object) X 2 (category consistency, different- vs. same-superordinate-category) design. Each condition consisted of 48 trials. On each trial, participants were presented with one image pair, with two colored markers displayed on both the source and target image. For each of the two colored markers, they were asked to 'move the marker on the top right corner in the target image to the corresponding location that maps to the same-color marker in the source image.' If the participant did not think there was an analogy between the two images, they were allowed to move the markers back to the top right corner. No time constraint was imposed. The entire experiment was completed in about 41 minutes on average. On each trial, the exact location of each marker placement was recorded.

**Analysis.** Since there are two markers on each trial, from the 41 participants, we collected 15,744 marker locations (41 participants × 192 problems × 2 markers), among which 2895 markers were excluded because the participants either did not move the marker or moved it back to the top right corner. In total, we have 12,849 marker locations for our analysis. Among the 2895 markers indicating the absence of an analogous part in the target images, 453 were from within-category problems, and 2442 were from cross-category problems.

The dependent measure in our experiment is the marker-offset-from-mean-location. We first computed the mean location (in 2D image space) for each marker in a problem by averaging reported marker locations across participants. We next calculated the Euclidean distance of each individual participant's marker displacement to the corresponding mean marker location. A smaller distance from the mean location indicates a typical marker location closer to the average; a larger distance indicates an atypical location far away from the mean placement. Hence, the dependent measure naturally captures human response variability in identifying the analogous part of the target image.

For different-superordinate-category problems, visual inspection of the pattern of marker placements across participants (see an example in Fig. 3b) indicated the potential presence of multiple clusters within the responses to a given problem. Therefore, we performed a dip test (with a threshold of $p < 0.05$) to identify the source markers for which the target markers had a bimodal distribution. For those markers, we then used the KMeans++ algorithm[55] to segregate responses into two clusters, and performed an additional analysis based on the average distance to the closest cluster mean.

### Reporting summary

Further information on research design is available in the Nature Portfolio Reporting Summary linked to this article.

## Data availability

Human behavioral data, model results, and original stimulus materials can be downloaded from: https://github.com/taylorwwebb/visiPAM.

## Code availability

Analysis code can be downloaded from: https://github.com/taylorwwebb/visiPAM. Model code is not accessible to the public due to the requirements of the funding agency (AFRL). However, it will be made available to individual members of the research community upon request. To request access to this code, please contact H.L.

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

## Acknowledgements

Preparation of this paper was supported by NSF Grant BCS-1827374 awarded to K.J.H. and AFRL grant FA8650-19-C-1692/S00017 to H.L. The views expressed in this paper are those of the authors and do not represent the views of any part of the US Government. This work was cleared for unlimited release under: AFRL-2023-3702.

## Author contributions

T.W., S.F., T.B., K.J.H., and H.L. designed the model and experiments. T.W. implemented the model simulations for 2D image mapping. S.F. implemented the model simulations for 3D object mapping. T.W. and S.F. performed data analysis for the human behavioral experiment. T.W., S.F., T.B., K.J.H., and H.L. drafted the manuscript.

## Competing interests

The authors declare no competing interests.
