## [Peer Review File · Nature Communications]

REVIEWER COMMENTS

Reviewer #1 (Remarks to the Author):

This is an interesting paper applying a previously published AI system to visual analogy problems--- finding corresponding object-part locations in pairs of images (both 2D and 3D). The paper will be of significance primarily to other researchers working computer vision, on "transfer learning" in ML, and on hybrid neural-symbolic approaches to analogy. The methodology is sound, and there is enough information in the paper to replicate the results.

My specific comments and suggestions are given below.

1. The visiPAM architecture is separated into a vision module and a mapping module, with no feedback between them. Thus the architecture seems wholly bottom-up --- a visual representation is constructed, and then fed into the mapping module. It would be useful to comment on whether feedback from the mapping module to the vision module---in order to reinterpret the input data and/or make certain object-part embeddings more similar---could be used to improve performance, and whether such feedback is important in human perception of analogies.
2. The authors note that the experiments reported here only use spatial relations, but that the architecture could be extended to include semantic relations. It would be helpful to add some speculation on how this might be done, and what kinds of tasks would require this.
3. I was confused by the discussion of the prior $p(M)$ that "favors isomorphic (one-to-one) mappings". What does this mean, exactly? My understanding is that the system is designed to do one-to-one mappings between pairs of nodes in the source and target graphs, so why is such a prior needed? Maybe I'm missing something here.
4. The authors have a very limited discussion of the kinds of errors made by their system. It would be helpful to include a more detailed analysis of the kinds of errors made, and thus give readers a better idea of the limitations of the current system.
5. In the section on 2D image mappings, for the between-category mappings of animals, the authors state: "We found that visiPAM performed comparably well in these task settings" -- comparably well with what?

6. In the section on 3D mapping, the authors state: "To compare the model's performance with human responses we applied the model to all image pairs used in the experiment, and measured the distance from the marker location predicted by visiPAM to the mean locations of human placements for each pair." How can this be a useful measure in cases where human placements are bimodal or have very high variance? It would be helpful if the authors could explain why this is a useful performance metric.

7. For the 3D mapping task: The authors state: "We showed that visiPAM ... very closely approached the overall level of human performance in this task." To me the word "performance" implies a kind of level of accuracy, but in the 3D task there are no correct answers, so what does "human performance" even mean? Maybe it would be clearer to say "human behavior" instead. This terminology is also used later on (p. 8).

9. At the end, the authors state, "We have presented a model capable of identifying abstract commonalities governing rich, high-dimensional visual inputs." This seems to be an overstatement of the actual results. In the 2D task, the accuracy is only about 60-70%, so the system is not able to identify such commonalities very robustly. In the 3D task, the quality of the results are even less clear. Also, the commonalities (e.g., mapping a cat leg to a horse leg or an airplane wing to another airplane wing) don't seem very *abstract* to me.

10. The authors state that visiPAM is "the first model capable of performing analogical mapping with real-world visual inputs". The authors should more carefully qualify this statement, since there are many papers in the ML literature about real-world image analogies, e.g., Sadeghi et al. 2015 ("Visalogy"); Peyre et al. 2019 ("Detecting unseen visual relations using analogies"), Wang et al. 2016 ("Contextual visual similarity"), Liao et al., 2017 ("Visual attribute transfer through deep image analogy"), Ichien et al, 2021 ("visual analogy: Deep learning versus compositional models"), and many others.

Reviewer #2 (Remarks to the Author):

1. Summary of the submission: The paper presents an effective approach (named visiPAM) for one shot part labeling task by combining iBot, DGCNN and PAM. By the combined benefit of each component, the proposed method outperforms the state of the art, SSMN, in Animals part-matching evaluation. The proposed visiPAM is shown to perform well in both 2D image and 3D point cloud domains.

2. Strengths of the proposed method

- Outperforming SSMN by large margin in 2D domain

- Easily extended to 3D domain and shows compelling results

3. Weaknesses of the proposed method

- Not sufficient novelty: The proposed visiPAM is an application of PAM for vision task using the well known advanced vision encoding models of iBot and DGCNN. Moreover, they are simply combined in a pipeline not even trained in an end-to-end manner which may result in a new combined learning model for such task.

- Lack of ablation study: Gains are likely coming from adopting an advanced vision encoding module such as iBot or an advanced point cloud encoding module such as DGCNN.

- No detailed analysis: The paper lacks the analysis of successful examples and failed examples but outlines the overall performance. It is less clear about the benefit that the proposed method can bring in solving this task.

- Much worse than human performance: As shown in Fig. 5, the proposed visiPAM performs much worse than the humans.

Reviewer #3 (Remarks to the Author):

The authors consider mapping locations between visual images and 3D point clouds. For example, visiPam can map the head of a cat in one image to the head of a horse in another image. visiPam relies on a novel pairing of existing algorithms. The first stage is to extract node representations. Different approaches are used for 2D and 3D inputs. Edges then are defined by distance. PAM then does the node mapping based on a combination of satisfying node and edge similarity constraints. PAM is described as Bayesian, though it may be understood as performing graph matching by minimizing a cost function involving these two terms.

The problem the authors consider is interesting and seems under explored. They cite and compare to one competing approach in the field. visiPam does well compared to that algorithm but seems to perform moderately in absolute terms. Admittedly, it's hard to say what constitutes good or bad performance in this application. One concern is that performance is largely driven by node matching. The authors do establish that edges lead to a significant performance gain, but the increment is modest. Likewise, correspondence with human performance was modest and appears largely driven by node matching.

One strength was that the approach builds upon previous work with PAM, though PAM itself appears to be a somewhat standard graph matching algorithm that captures key ideas in analogy. Perhaps an important part of the contribution is that aspects of visual reasoning can be addressed by combining

these basic components. The approach is sensible and it's refreshing to encounter analogy research involving image-computable models.

For 2D images, edges were determined by distance measures in the 2D plane. This seems like it could lead to large errors depending on how objects were positioned in 3D space? It was unclear to me what role the two distance measures played. Could the authors provide some intuition or even numerical results that informed this choice?

For the 2D case, why was the between category performance for the animals higher than within for visIPam (Table 2)? This result seems to conflict with the result shown in Figure 5.

It appears to be a limitation of the datasets used (e.g., Pascal Part Matching) that the number of classes compared is very small. Somewhat concerning, there appears to be differences between classes (e.g., Figure 5). On the positive side, there are a lot of cases contained within these classes.

How does orientation, eccentricity, etc. affect performance? For example, if an image is reflected, is performance affected? Does the model suffer relative to humans when the cat in one image is facing the opposite direction as the cat in the other image? Is the model limited to canonical orientations? More generally, the model works with images so I was expecting more consideration of how image properties affect performance with an eye toward identifying any shortcuts the model may be using. I.e., it looks like it's reasoning but is actually exploiting lower-level image information or confounds.

Is there good evidence for the model and people that mappings between images are systematic? For example, when the model gets a mapping correct, are the neighboring mappings more likely to be correct? This is related to the authors' discussion of bimodal solutions.

One minor suggestion is to move Figure 1 into the Introduction. I didn't really get the basic idea of the model until digesting Figure 1.

To conclude, I found this work interesting and wish the authors continued success.

We would like to thank the reviewers for their constructive feedback on our work. We have now revised the manuscript to address these issues. In particular, we have performed additional ablation experiments and error analyses to better understand visiPAM's strengths and limitations. We have also revised the text so as to more carefully characterize visiPAM's contribution relative to prior work, and have added a number of clarifications about the methodology and results. We believe that the revised manuscript is significantly improved thanks to the reviewers' many insightful comments and suggestions.

Below we include a point-by-point reply to the issues raised by the reviewers, along with a description of the corresponding revisions made to the manuscript. The reviewers' comments are presented in blue, and revisions are presented in red.

Reviewer 1:

1. The visiPAM architecture is separated into a vision module and a mapping module, with no feedback between them. Thus the architecture seems wholly bottom-up --- a visual representation is constructed, and then fed into the mapping module. It would be useful to comment on whether feedback from the mapping module to the vision module---in order to reinterpret the input data and/or make certain object-part embeddings more similar---could be used to improve performance, and whether such feedback is important in human perception of analogies.

This is an excellent suggestion for future model development, which we now raise in the final paragraph of Discussion:

Finally, an important next step will be to expand the capabilities of visiPAM to incorporate other basic cognitive elements of analogy. For example, human analogical reasoning often involves an interaction between bottom-up and top-down processes, in which the reasoning process can sometimes guide the reinterpretation of perceptual inputs [5, 8, 42]. VisiPAM in its current formulation is purely bottom-up – visual representations are constructed by the vision module, and then passed to PAM to perform mapping – but a promising avenue for future work will be to incorporate top-down feedback to the vision module. One possibility for doing so is to incorporate additional parameters (e.g., parameters for visual feature attention) to visiPAM's optimization procedure, such that the visual representations are modulated based on the downstream graph-matching objective. Additionally, the present work focused on the identification of analogies between pairs of images, but human reasoners are also capable of inducing more general schematic representations that capture abstract similarity across multiple examples [43]. Future work should investigate how visiPAM's reasoning mechanisms can be extended to capture this capacity for schema induction. There are many exciting prospects for the continued synergy between rich visual representations and human-like reasoning mechanisms.

[5] D. R. Hofstadter, M. Mitchell, et al., "The copycat project: A model of mental fluidity and analogy-making," *Advances in connectionist and neural computation theory*, vol. 2, pp. 205–267, 1995.

[8] D. J. Chalmers, R. M. French, and D. R. Hofstadter, "High-level perception, representation, and analogy: A critique of artificial intelligence methodology," *Journal of Experimental & Theoretical Artificial Intelligence*, vol. 4, no. 3, pp. 185–211, 1992.

[42] M. Mitchell, *Analogy-making as perception: A computer model*. MIT Press, 1993.

[43] M. L. Gick and K. J. Holyoak, "Schema induction and analogical transfer," *Cognitive psychology*, vol. 15, no. 1, pp. 1–38, 1983.

2. The authors note that the experiments reported here only use spatial relations, but that the architecture could be extended to include semantic relations. It would be helpful to add some speculation on how this might be done, and what kinds of tasks would require this.

We have added the following in the penultimate paragraph of Discussion (new part is in bold):

One limitation of the present work is the lack of a method for deriving 3D representations (such as point-clouds) from the 2D inputs that human reasoners typically receive. **We found the incorporation of topological information (e.g., whether two parts are connected) significantly improved visiPAM's mapping performance, but it would be desirable to derive this information directly from 2D inputs.** Some recent advances may make it possible to take a step in that direction [39], but visiPAM will likely benefit from the rapid pace of innovation in representation learning algorithms. We are particularly excited about the continued development of general-purpose, self-supervised algorithms (such as the iBOT algorithm that we employed [21]), which we hope will provide increasingly rich representations over which reasoning algorithms such as PAM can operate. **Similarly, it will be useful in future work to incorporate non-visual semantic and functional information into visiPAM's representations. We found that this kind of information was a significant contributing factor to human mapping judgments. Potential sources of such knowledge include word embeddings [40], multimodal embeddings [41], and representations of relations between verbal concepts [26].**

[21] J. Zhou et al., "Ibot: Image bert pre-training with online tokenizer," in *10th International Conference on Learning Representations, ICLR, 2022*.

[26] H. Lu, Y. N. Wu, and K. J. Holyoak, "Emergence of analogy from relation learning," *Proceedings of the National Academy of Sciences*, vol. 116, no. 10, pp. 4176–4181, 2019.

[39] R. Ranftl, A. Bochkovskiy, and V. Koltun, "Vision transformers for dense prediction," in *Proceedings of the IEEE/CVF International Conference on Computer Vision, 2021*, pp. 12 179–12 188.

[40] T. Mikolov, I. Sutskever, K. Chen, G. S. Corrado, and J. Dean, "Distributed representations of words and phrases and their compositionality," *Advances in neural information processing systems*, vol. 26, 2013.

[41] A. Radford et al., "Learning transferable visual models from natural language supervision," in *International conference on machine learning, PMLR, 2021*, pp. 8748–8763.

Please note also that we have included some examples of the kinds of problems for which human responses seem to incorporate semantic information. These are described in the error analyses reported below (under comment #4).

3. I was confused by the discussion of the prior $p(\mathbf{M})$ that "favors isomorphic (one-to-one) mappings". What does this mean, exactly? My understanding is that the system is designed to do one-to-one mappings between pairs of nodes in the source and target graphs, so why is such a prior needed? Maybe I'm missing something here.

We have clarified this point in the end of final paragraph in Results, which describes the model:

In addition, PAM incorporates a prior $p(\mathbf{M})$ that favors isomorphic (one-to-one) mappings. In other words, a mapping function with one-to-one correspondences between the source and the target is assigned a higher prior probability. At the algorithm level, given that an exhaustive search over possible mappings was not feasible (we address problems with up to 10 object parts, corresponding to $10! \approx 3.6$ million possible one-to-one mappings), we used a graduated assignment algorithm [23]. **In this approach, \mathbf{M} is initialized as a uniform mapping, in which each source node is equally likely to map to any target node, and this mapping is then iteratively updated based on the likelihood in Equation 2. This iterative procedure is also guided by the prior $p(\mathbf{M})$ to encourage convergence to an isomorphic (one-to-one) mapping.**

4. The authors have a very limited discussion of the kinds of errors made by their system. It would be helpful to include a more detailed analysis of the kinds of errors made, and thus give readers a better idea of the limitations of the current system.

This is an excellent suggestion. We have now added more detailed analyses of errors, reported in expanded final two paragraphs of section 2.2 (results for 2D image mapping):

Figure 2 shows some examples of the mappings produced by visiPAM. These include some impressive successes, including mapping of a large number of object parts across dramatic differences in background, lighting, pose, and visual appearance (Figure 2a), as well as between objects of different categories (Figure 2b). **Notably, we also found that visiPAM's performance was unaffected when the target image was horizontally reflected relative to the source (Supplementary Section S1.1 and Table S1), suggesting that the model is robust to low-level image manipulations.**

Our analyses also illustrated some of the model's limitations. Figure 2c shows an example of an error pattern that we commonly observed, in which visiPAM confused corresponding left and right parts (in this case the left and right feet in a comparison of two horses). **A more comprehensive analysis is shown in Supplementary Figures S1-S3, where it can be seen that this kind of confusion of corresponding lateralized parts accounts for a large percentage of visiPAM's errors.** We also found that visiPAM particularly struggled with mapping images of planes (achieving the model's lowest within-category mapping accuracy of 42.5%), as shown in Figure 2d. This is likely due to the limitation of the 2D spatial relations used in this version of the model, which are particularly ill-suited to objects such as planes that can appear in a wide variety of poses and viewpoints. Thus, while visiPAM shows an impressive ability to perform mappings between complex, real-world images using only these 2D spatial

relations, we suspect that accurate 3D spatial knowledge is likely necessary to solve some challenging visual analogy problems at a human level. To address this concern, we next sought to evaluate the performance of visiPAM in the context of 3D visual inputs.

The corresponding supplementary results section is as follows:

S1 Error analysis

S1.1 2d image mapping

We conducted additional analyses to better understand visiPAM's strengths and limitations in the context of the 2D image mapping task. First, we performed a test in which the target image was horizontally reflected relative to the source image. This experiment allowed us to test the extent to which visiPAM's performance might depend on canonical orientation. We found that this manipulation did not affect visiPAM's performance for either within-category or between-category animal mapping (Table S1), suggesting that the model is robust to these kinds of low-level image manipulations.

	Within-category	Between-category
	Animals	
VisiPAM (default)	63.2% (59.1%)	67.9% (59.9%)
VisiPAM (target horizontally reflected)	62.7% (58.6%)	67.9% (59.9%)
Random	10%	20%

Table S1: VisiPAM's performance is robust to image manipulations. Mapping accuracy for both within-category (e.g., mapping from cat to cat, or horse to horse) and between-category (e.g., mapping from cat to horse) animal comparisons was largely unaffected by horizontal reflection of the target image. 'Random' denotes chance performance (determined by average number of part comparisons in each condition). Values in parentheses reflect chance-normalized performance (percentage of the range between chance performance and 100% accuracy).

We also evaluated the frequency with which visiPAM makes specific types of errors. Specifically, we computed the frequency with which each source part was mapped to each target part. The results of this analysis for both within-category and between-category animal comparisons are shown in Figures S1-S3. The most common type of error involved confusion of corresponding lateralized parts (e.g., confusion of left and right ears, or left and right legs). There were also occasionally errors related to spatial proximity of parts (e.g., confusion of the neck and head).

Figure S1: Error pattern for within-category animal mapping – cat to cat. Mapping frequency for each pair of source (rows) and target (column) parts. Perfect performance would correspond to a mapping frequency of 1 for all entries along the diagonal.

Figure S2: Error pattern for within-category animal mapping – horse to horse.

Figure S3: Error pattern for between-category animal mapping – cat to horse.

We have also added an additional error analysis for the 3D mapping task in Supplementary Materials (and added a pointer in the corresponding section of the main paper):

S1.2 3D object mapping

We conducted a qualitative analysis of the errors generated by visiPAM in the context of the 3D marker mapping experiment. We identified two primary causes of errors in predicting human marker placements: (1) the model's over-reliance on local 3D geometric properties (e.g., curvature, corner, T-junctions); and (2) a lack of functional knowledge or semantic information pertaining to the parts. Figure S4 illustrates a few examples in which visiPAM gives undue attention to local geometric structure in its mapping decisions. Circles indicate visiPAM's predicted mapping locations, and crosses indicate the mean location of human responses. For example, for the image pairs in Figure S4a, visiPAM maps the green marker on the back of the elephant (source) to a position (green circle) located on the lower back of the ox (target). For this mapping, the model largely relies on local geometric commonalities due to their similar surface curvature. Humans, on the other hand, map the green dot to the middle of the ox's spine based on knowledge of overall body structure. In Figures S4b and S4c, the model maps the green marker using local geometric cues of the corner (L junction). In contrast, humans use functional knowledge to select the back leg of the target object (green cross), which provides support for the object. The mapping of chairs in Figure S4d shows a similar mapping error (red circles) based on shared vertical surfaces, whereas humans appear to rely on semantic knowledge to match the "back" of the chair to the "back" of the horse. Similarly, in Figure S4e, the model maps the green marker on the basis of shared flat surfaces (e.g., seating surface of a chair, and flat belly of an ox), whereas humans are sensitive to the functional support relations. Lastly, Figure S4f shows an interesting case in which the model maps the front leg of the elephant (red marker) to a horizontal bar of the chair. In this instance, the model matches the

back legs of the elephant to the chair’s vertical legs; accordingly, because the model favors one-to-one mapping, the two front legs of the elephant are mapped to the remaining narrow surfaces (despite the mismatch of orientations).

Figure S4: Examples of errors made by visiPAM on 3D mapping task. In each panel, images on the left panel are source images, and images on the right are target images that show both visiPAM predictions (circle) and human response centers (cross).

5. In the section on 2D image mappings, for the between-category mappings of animals, the authors state: "We found that visiPAM performed comparably well in these task settings" -- comparably well with what?

This has been revised:

We found that visiPAM performed comparably well across all three of these conditions (within- and between-category animal mapping, within-category vehicle mapping).

Note also that these conditions involve different numbers of part comparisons, meaning that chance performance is different in each condition. We have revised Table 1 to make this more explicit:

	Within-category		Between-category
	Animals	Vehicles	Animals
VisiPAM	63.2% (59.1%)	69.5% (58.8%)	67.9% (59.9%)
VisiPAM (nodes only)	55.5% (50.6%)	61.6% (48.1%)	62.3% (52.9%)
VisiPAM (edges only)	47.8% (42%)	63.7% (50.9%)	51.5% (39.4%)
SSMN	46.6% (40.7%)		
Random	10%	26%	20%

Table 1: Analogical mapping with 2D images. Mapping accuracy on part-matching task proposed by Choi et al. [23]. Original task in [23] involved evaluation on within-category animal comparisons after training with 37,330 mapping problems. VisiPAM significantly outperformed SSMN, the previous state-of-the-art, despite having no direct training on mapping. VisiPAM also performed well on new problems involving within-category vehicle comparisons, and between-category animal comparisons (e.g., mapping from cat to horse). VisiPAM performed best when mapping was based on both node and edge similarity. 'Random' denotes chance performance (determined by average number of part comparisons). **Values in parentheses reflect chance-normalized performance (percentage of the range between chance performance and 100% accuracy). VisiPAM's performance is roughly comparable across all conditions once chance performance is taken into account.**

6. In the section on 3D mapping, the authors state: "To compare the model's performance with human responses we applied the model to all image pairs used in the experiment, and measured the distance from the marker location predicted by visiPAM to the mean locations of human placements for each pair." How can this be a useful measure in cases where human placements are bimodal or have very high variance? It would be helpful if the authors could explain why this is a useful performance metric.

We chose to use distance to the overall mean of human responses in each condition because we wanted a single measure that captured both the increased ambiguity in the different-category condition (leading to bimodal responses for some problems) and the increased variability. However, we agree that it is important to also determine whether the increased variability is only a function of the fact that responses were bimodal for some problems. To address this issue, we performed an additional analysis in which each problem was classified as having either a unimodal or bimodal human response distribution (using a dip test for bimodality). For problems that were classified as having a bimodal response distribution, we quantified responses based on the distance to the mean of the closest cluster (out of two). For problems that were classified as having a unimodal response distribution, we quantified responses based on the distance to the overall mean, as in the primary analysis. This analysis revealed the same qualitative effects for both human responses and visiPAM as seen in the primary analysis.

The following text has been added to the end of the fourth paragraph of section 2.3:

The same qualitative results were present when taking the bimodal nature of human responses into account (i.e., when measuring distance to the mean of the closest human cluster for trials with a bimodal response distribution), though both visiPAM and human participants showed lower variance when behavior was quantified in this manner (Supplementary Figure S7).

The corresponding supplementary section is as follows:

s3 Cluster analysis for 3D object mapping

Figure S7: Comparison of visiPAM with human behavior when controlling for bimodal response distribution. Violin plot comparing human placements and visiPAM predictions. Each problem was classified as having either a unimodal or bimodal response distribution (for human responses) using a dip test (with a threshold of $p < 0.05$). For problems with a unimodal distribution, responses were quantified based on the distance to the overall mean for that trial (as in the primary results displayed in Figure 5). For problems with a bimodal distribution, a clustering analysis was performed to sort responses into two clusters, and responses were quantified based on the distance to the mean of the closest cluster. This was done for both visiPAM and human responses. All problems in the same-superordinate-category condition were classified as unimodal (thus the results in this condition are identical to Figure 5). The dip test classified 48.4% of problems in the different-superordinate-category condition as bimodal. Even after accounting for this bimodality, responses were still more variable in the different-superordinate-category condition (for both humans and visiPAM). Strip plots (small colored dots) indicate average distance in each condition, one point for each participant. Thin gray lines show the within-participant differences for the same- vs. different-superordinate-category conditions. Violin plots exclude data points greater than 2.5 standard deviations away from mean, though these individual data points are included in strip plots. Black horizontal lines indicate mean human distances in each condition, and error bars indicate standard deviations. Large green dots indicate visiPAM predictions (i.e., the distance between model predicted location and human mean location).

7. For the 3D mapping task: The authors state: "We showed that visiPAM ... very closely approached the overall level of human performance in this task." To me the word "performance" implies a kind of level of accuracy, but in the 3D task there are no correct answers, so what does "human performance" even mean? Maybe it would be clearer to say "human behavior" instead. This terminology is also used later on (p. 8).

We agree that this is a more precise term. We have made the following changes to address this:

We revised the penultimate sentence of the introduction as follows:

We also performed a human experiment involving visual analogies between 3D objects from disparate categories (e.g., an analogy between an animal and a man-made object), where we found that visiPAM closely matched the pattern of human **behavior variability** across conditions; ~~and approached the level of human performance.~~

The beginning of the fourth paragraph of section 2.3 has been revised as follows:

To systematically compare the model's ~~performance~~ **behavior** with human responses, we applied the model to all image pairs used in the experiment, and measured the distance from the marker location predicted by visiPAM to the mean locations of human placements for each pair. Figure 5 shows the results of this comparison. We found that visiPAM ~~both~~ reproduced the qualitative pattern displayed by human mappings across conditions; ~~and also very closely approached the level of human performance in this task.~~

The final sentence of the first paragraph of the Discussion has been revised as follows:

Our experiment with mapping of 3D object parts showed that, when armed with rich 3D visual representations, visiPAM ~~both approached the overall level of human performance, and~~ matched the pattern of human mappings across conditions.

9. At the end, the authors state, "We have presented a model capable of identifying abstract commonalities governing rich, high-dimensional visual inputs." This seems to be an overstatement of the actual results. In the 2D task, the accuracy is only about 60-70%, so the system is not able to identify such commonalities very robustly. In the 3D task, the quality of the results are even less clear. Also, the commonalities (e.g., mapping a cat leg to a horse leg or an airplane wing to another airplane wing) don't seem very *abstract* to me.

Although we agree that not all comparisons were abstract, some of them (e.g., mapping between animals and chairs) seem fairly abstract. For these far analogy problems, visiPAM's predicted mapping locations were within 1 standard deviation of human responses. In any case, we have revised the sentence to tone down our claim:

We have presented a computational model that performs zero-shot analogical mapping based on rich, high-dimensional visual inputs.

10. The authors state that visiPAM is "the first model capable of performing analogical mapping with real-world visual inputs". The authors should more carefully qualify this statement, since there are many papers in the ML literature about real-world image analogies, e.g., Sadeghi et al. 2015 ("Visalogy"); Peyre et al. 2019 ("Detecting unseen visual relations using analogies"), Wang et al. 2016 ("Contextual visual similarity"), Liao et al., 2017 ("Visual attribute transfer through deep image analogy"), Ichien et al, 2021 ("visual analogy: Deep learning versus compositional models"), and many others.

In order to clarify the connections with previous work, this paragraph has been revised as follows:

These results build on other recent work that has employed similarity-based reasoning over representations derived using deep learning [20, 26, 27], enabling analogy to emerge zero-shot. Relative to this previous work, visiPAM is unique in that it both operates over real-world visual inputs, and involves analogical mapping (rather than computing similarity of pre-mapped components). VisiPAM also has commonalities with some recent deep learning models. These include models of visual analogy that are applied to real-world visual inputs [24, 28–30], and models that incorporate graph-based representations similar to visiPAM's structured visual representations [31–33]. Unlike visiPAM, these models all involve end-to-end training on a large number of examples for a specific task, with limited generalization to new analogy tasks.

[20] H. Lu, N. Ichien, and K. J. Holyoak, "Probabilistic analogical mapping with semantic relation networks," *Psychological Review*, 2022.

[24] J. Choi, J. Krishnamurthy, A. Kembhavi, and A. Farhadi, "Structured set matching networks for one-shot part labeling," in *Proceedings of the IEEE Conference on Computer Vision and Pattern Recognition*, 2018, pp. 3627–3636.

[26] H. Lu, Y. N. Wu, and K. J. Holyoak, "Emergence of analogy from relation learning," *Proceedings of the National Academy of Sciences*, vol. 116, no. 10, pp. 4176–4181, 2019.

[27] N. Ichien, Q. Liu, S. Fu, K. Holyoak, A. Yuille, and H. Lu, "Visual analogy: Deep learning versus compositional models," in *Proceedings of the 43rd Annual Meeting of the Cognitive Science Society*, 2021.

[28] F. Sadeghi, C. L. Zitnick, and A. Farhadi, "Visalogy: Answering visual analogy questions," *Advances in Neural Information Processing Systems*, vol. 28, 2015.

[29] J. Liao, Y. Yao, L. Yuan, G. Hua, and S. B. Kang, "Visual attribute transfer through deep image analogy," *arXiv preprint arXiv:1705.01088*, 2017.

[30] J. Peyre, I. Laptev, C. Schmid, and J. Sivic, "Detecting unseen visual relations using analogies," in *Proceedings of the IEEE/CVF International Conference on Computer Vision*, 2019, pp. 1981–1990.

[31] A. Santoro et al., "A simple neural network module for relational reasoning," *Advances in neural information processing systems*, vol. 30, 2017.

[32] P. W. Battaglia et al., "Relational inductive biases, deep learning, and graph networks," *arXiv preprint arXiv:1806.01261*, 2018.

[33] T. N. Kipf, E. van der Pol, and M. Welling, "Contrastive learning of structured world models," in *8th International Conference on Learning Representations, ICLR*, 2020.

Reviewer 2:

3. Weaknesses of the proposed method

- Not sufficient novelty: The proposed visiPAM is an application of PAM for vision task using the well known advanced vision encoding models of iBot and DGCNN. Moreover, they are simply combined in a pipeline not even trained in an end-to-end manner which may result in a new combined learning model for such task.

We thank the reviewer for this suggestion. We agree that it would be interesting to train our model in an end-to-end manner, as this may be a promising direction for some computer vision applications. In principle, this should be relatively straightforward, given that the model is already fully differentiable. However, in the present work, our objective was to model the human capacity for zero-shot analogical reasoning – that is, the ability to make analogies without direct training on specific analogy tasks. For this reason, we specifically avoided end-to-end training directly on our mapping task, and instead used representations learned via more general purpose approaches (e.g., self-supervised learning).

We agree that it would be interesting in future work to explore a version of the model that is trained end-to-end. We have added a note to the description of the model in section 2.1 (Computational framework) highlighting this possibility:

It should be noted that this approach is fully differentiable, and therefore the entire model, including the vision module, could in principle be learned end-to-end based on the mapping task. However, since our goal in the present work was to account for the human capacity for zero-shot analogical reasoning, we instead use pretrained representations, such that no direct training on the mapping task is required.

- Lack of ablation study: Gains are likely coming from adopting an advanced vision encoding module such as iBot or an advanced point cloud encoding module such as DGCNN.

The reviewer is correct that effective visual embeddings are one important aspect of our proposed model. However, our ablation analyses indicate that other aspects of the model also play an important role, most notably the edge embeddings. To assess the importance of the node embeddings (from either iBOT or DGCNN) or edge embeddings (spatial relations), we performed ablation experiments in which mapping was based only on nodes, or only on edges. The results clearly demonstrate that both components are important – visiPAM performs worse both when using nodes only, and when using edges only. Therefore, it is not the case that the gains demonstrated by our model stem exclusively from the use of advanced vision encoding methods. The edge embeddings used by the model are also an important contributor to its performance.

The following excerpt from section 2.2 of the paper describes these ablation experiments for the 2D image mapping task:

A key element of our proposed model is that it performs mapping based on the similarity of both object parts and their relations. To determine the importance of this design decision, we performed ablations on either node or edge similarity components, by setting α to either 0 or 1. We found that both of these ablations significantly impaired the performance of visiPAM. This pattern aligns with findings from studies of human analogy-making, which show that human reasoners are typically sensitive to similarity of both entities and relations [25].

	Within-category		Between-category
	Animals	Vehicles	Animals
VisiPAM	63.2% (59.1%)	69.5% (58.8%)	67.9% (59.9%)
VisiPAM (nodes only)	55.5% (50.6%)	61.6% (48.1%)	62.3% (52.9%)
VisiPAM (edges only)	47.8% (42%)	63.7% (50.9%)	51.5% (39.4%)
SSMN	46.6% (40.7%)		
Random	10%	26%	20%

Table 1: Analogical mapping with 2D images. Mapping accuracy on part-matching task proposed by Choi et al. [23]. Original task in [23] involved evaluation on within-category animal comparisons after training with 37,330 mapping problems. VisiPAM significantly outperformed SSMN, the previous state-of-the-art, despite having no direct training on mapping. VisiPAM also performed well on new problems involving within-category vehicle comparisons, and between-category animal comparisons (e.g., mapping from cat to horse). VisiPAM performed best when mapping was based on both node and edge similarity. ‘Random’ denotes chance performance (determined by average number of part comparisons). Values in parentheses reflect chance-normalized performance (percentage of the range between chance performance and 100% accuracy). VisiPAM’s performance is roughly comparable across all conditions once chance performance is taken into account.

VisiPAM’s edge embeddings contained two distinct kinds of spatial relations, one based on angular distance (r_θ) and one based on vector difference (r_δ). We performed an additional ablation experiment that tested the importance of these different components. The results, which we have now included in the Supplementary Material, demonstrated that both of these types of spatial relation contributed to visiPAM’s performance:

S4 Ablation analyses

S4.1 2D image mapping

VisiPAM’s edge embeddings involved two different types of spatial relations, one based on angular distance (r_θ), and one based on vector difference (r_δ ; see Section 4.2.3 for more details). We performed ablation experiments in which visiPAM’s edge embeddings only contained one of these spatial relation types. The results demonstrated that both components contributed significantly to visiPAM’s performance (Table S2).

	Within-category		Between-category
	Animals	Vehicles	Animals
VisiPAM	63.2% (59.1%)	69.5% (58.8%)	67.9% (59.9%)
VisiPAM (r_θ only)	56.7% (51.8%)	66.7% (55%)	67.5% (59.4%)
VisiPAM (r_δ only)	56.1% (51.2%)	65.4% (53.2%)	52.5% (40.6%)
Random	10%	26%	20%

Table S2: Analogical mapping with 2D images – edge embedding ablation analyses. Mapping accuracy on part-matching task. VisiPAM performed worse when edge embeddings were based only on angular distance (r_θ) or vector difference (r_δ). ‘Random’ denotes chance performance (determined by average number of part comparisons). Values in parentheses reflect chance-normalized performance (percentage of the range between chance performance and 100% accuracy).

We also found that visiPAM’s performance can be even further improved by augmenting edges with information about whether two object parts are connected (based on ground-truth information). Though these experiments rely on ground-truth knowledge about object parts, the results suggest that the addition of new sources of relational information will likely lead to continued improvement. These results are also described in the Supplementary Material:

We also carried out an additional experiment to determine whether visiPAM’s performance can be further improved by incorporating additional sources of relational information. Specifically, we augmented visiPAM’s edge embeddings with information about whether two parts are connected. This is an important aspect of relations between object parts that may not be entirely captured by simple spatial relations, especially for non-rigid body objects such as animals. For instance, the spatial relation between two body parts (e.g., the hand and the shoulder) can change dramatically depending on posture, but the topological relationship between body parts (e.g., which body parts are connected) remains unchanged. To test whether this information might further improve visiPAM’s performance, we tested a model in which edges were modified based on whether the corresponding object parts were connected (using ground-truth knowledge). For parts that were not connected, the values for the spatial relation components (r_θ and r_δ) were set to 0, while spatial relation embeddings were left unchanged for connected parts. Additionally, an extra dimension was added to the edge embeddings with a value of 1 for connected parts, and a value of -1 for non-connected parts. We found that this version of visiPAM performed even better than the version that included spatial relations only, particularly for mapping problems involving animals (Table S3). Though this preliminary experiment relied on ground-truth knowledge about the object parts, future work might explore methods for extracting this information directly from images.

	Within-category		Between-category
	Animals	Vehicles	Animals
	72.4% (69.3%)	70.3% (59.9%)	77% (71.3%)

Table S3: VisiPAM’s performance is improved by incorporating topological relations. VisiPAM’s performance was improved when edges were augmented with information about whether two parts were connected. Values in parentheses reflect chance-normalized performance (percentage of the range between chance performance and 100% accuracy).

We also performed an ablation analysis for the 3D object mapping task. As with the 2D image task, we found that visiPAM performed worse (in the sense that it deviated more from human responses) when it was based on only node similarity or edge similarity. This further demonstrates that visiPAM’s performance is not driven solely by its visual embeddings (nodes), but also depends on its spatial relation embeddings (edges). These results have been added to the Supplementary Material:

	Same superordinate category		Different superordinate category		r
	Animal-to-animal	Chair-to chair	Animal-to-chair	Chair-to-animal	
VisiPAM	11.51	17.03	34.37	36.63	0.70
VisiPAM (nodes only)	13.77	19.96	35.83	37.98	0.61
VisiPAM (edges only)	12.61	19.32	34.86	40.10	0.60

Table S4: Ablation results for comparison of visiPAM with human behavior. VisiPAM performed better than ablated models based on either node or edge similarity only. Results in the middle four columns reflect average distance between visiPAM and human marker placements (smaller values indicate better fit to human responses) in each of four conditions, defined by the target category (animal vs. chair), and whether or not the source and target objects were from the same (e.g., animal-to-animal) or different (e.g., animal-to-chair) superordinate categories. Results in the rightmost column reflect the item-level correlation between human and visiPAM distances to the human mean placement (higher values indicate a better fit to human responses).

- No detailed analysis: The paper lacks the analysis of successful examples and failed examples but outlines the overall performance. It is less clear about the benefit that the proposed method can bring in solving this task.

This is an excellent suggestion. We have now added additional error analyses to better understand visiPAM’s strengths and limitations. Specifically, we have expanded final two paragraphs of section 2.2 (results for 2D image mapping):

Figure 2 shows some examples of the mappings produced by visiPAM. These include some impressive successes, including mapping of a large number of object parts across dramatic differences in background, lighting, pose, and visual appearance (Figure 2a), as well as between objects of different categories (Figure 2b). **Notably, we also found that visiPAM’s performance was unaffected when the target image was horizontally reflected relative to the source (Supplementary Section S1.1 and Table S1), suggesting that the model is robust to low-level image manipulations.**

Our analyses also illustrated some of the model’s limitations. Figure 2c shows an example of an error pattern that we commonly observed, in which visiPAM confused corresponding left and right parts (in this case the left and right feet in a comparison of two horses). **A more**

comprehensive analysis is shown in Supplementary Figures S1-S3, where it can be seen that this kind of confusion of corresponding lateralized parts accounts for a large percentage of visiPAM’s errors. We also found that visiPAM particularly struggled with mapping images of planes (achieving the model’s lowest within-category mapping accuracy of 42.5%), as shown in Figure 2d. This is likely due to the limitation of the 2D spatial relations used in this version of the model, which are particularly ill-suited to objects such as planes that can appear in a wide variety of poses and viewpoints. Thus, while visiPAM shows an impressive ability to perform mappings between complex, real-world images using only these 2D spatial relations, we suspect that accurate 3D spatial knowledge is likely necessary to solve some challenging visual analogy problems at a human level. To address this concern, we next sought to evaluate the performance of visiPAM in the context of 3D visual inputs.

The corresponding supplementary results section is as follows:

S1 Error analysis

S1.1 2d image mapping

We conducted additional analyses to better understand visiPAM’s strengths and limitations in the context of the 2D image mapping task. First, we performed a test in which the target image was horizontally reflected relative to the source image. This experiment allowed us to test the extent to which visiPAM’s performance might depend on canonical orientation. We found that this manipulation did not affect visiPAM’s performance for either within-category or between-category animal mapping (Table S1), suggesting that the model is robust to these kinds of low-level image manipulations.

	Within-category	Between-category
	Animals	
VisiPAM (default)	63.2% (59.1%)	67.9% (59.9%)
VisiPAM (target horizontally reflected)	62.7% (58.6%)	67.9% (59.9%)
Random	10%	20%

Table S1: VisiPAM’s performance is robust to image manipulations. Mapping accuracy for both within-category (e.g., mapping from cat to cat, or horse to horse) and between-category (e.g., mapping from cat to horse) animal comparisons was largely unaffected by horizontal reflection of the target image. ‘Random’ denotes chance performance (determined by average number of part comparisons in each condition). Values in parentheses reflect chance-normalized performance (percentage of the range between chance performance and 100% accuracy).

We also evaluated the frequency with which visiPAM makes specific types of errors. Specifically, we computed the frequency with which each source part was mapped to each target part. The results of this analysis for both within-category and between-category animal comparisons are shown in Figures S1-S3. The most common type of error involved confusion of corresponding lateralized parts (e.g., confusion of left and right ears, or left and right legs). There were also occasionally errors related to spatial proximity of parts (e.g., confusion of the neck and head).

Figure S1: Error pattern for within-category animal mapping – cat to cat. Mapping frequency for each pair of source (rows) and target (column) parts. Perfect performance would correspond to a mapping frequency of 1 for all entries along the diagonal.

Figure S2: Error pattern for within-category animal mapping – horse to horse.

Figure S3: Error pattern for between-category animal mapping – cat to horse.

We have also added additional error analysis for the 3D mapping task in Supplementary Materials (and added a pointer to the corresponding section of the main paper):

S1.2 3D object mapping

We conducted a qualitative analysis of the errors generated by visiPAM in the context of the 3D marker mapping experiment. We identified two primary causes of errors in predicting human marker placements: (1) the model's over-reliance on local 3D geometric properties (e.g., curvature, corner, T-junctions); and (2) a lack of functional knowledge or semantic information pertaining to the parts. Figure S4 illustrates a few examples in which visiPAM gives undue attention to local geometric structure in its mapping decisions. Circles indicate visiPAM's predicted mapping locations, and crosses indicate the mean location of human responses. For example, for the image pairs in Figure S4a, visiPAM maps the green marker on the back of the elephant (source) to a position (green circle) located on the lower back of the ox (target). For this mapping, the model largely relies on local geometric commonalities due to their similar surface curvature. Humans, on the other hand, map the green dot to the middle of the ox's spine based on knowledge of overall body structure. In Figures S4b and S4c, the model maps the green marker using local geometric cues of the corner (L junction). In contrast, humans use functional knowledge to select the back leg of the target object (green cross), which provides support for the object. The mapping of chairs in Figure S4d shows a similar mapping error (red circles) based on shared vertical surfaces, whereas humans appear to rely on semantic knowledge to match the "back" of the chair to the "back" of the horse. Similarly, in Figure S4e, the model maps the green marker on the basis of shared flat surfaces (e.g., seating surface of a chair, and flat belly of an ox), whereas humans are sensitive to the functional support relations. Lastly, Figure S4f shows an interesting case in which the model maps the front leg of the elephant (red marker) to a horizontal bar of the chair. In this instance, the model matches the

back legs of the elephant to the chair's vertical legs; accordingly, because the model favors one-to-one mapping, the two front legs of the elephant are mapped to the remaining narrow surfaces (despite the mismatch of orientations).

Figure S4: Examples of errors made by visiPAM on 3D mapping task. In each panel, images on the left panel are source images, and images on the right are target images that show both visiPAM predictions (circle) and human response centers (cross).

- Much worse than human performance: As shown in Fig. 5, the proposed visiPAM performs much worse than the humans.

We would like to emphasize that visiPAM produced very similar responses to human participants in the 3D object mapping task. VisiPAM's responses were an average of 25 pixels away from the mean human placement, whereas human responses were an average of 20 pixels from the mean placement. Relative to the size of the objects (average height of 213 pixels and width of 135 pixels), an average difference of 5 pixels is not very large. VisiPAM also showed a reasonably large correlation with human responses at the individual item level ($r = 0.7$ across 384 analogical mapping judgements).

Note also that we have modified the implementation details for 3D object mapping to make the model more consistent with the implementation used for 2D image mapping. This includes the use of bistochastic normalization prior to performing mapping, employing a different value for the α , the parameter that governs the relative influence of node and edge similarity ($\alpha=0.9$), and using edge embeddings based on both angular and vector difference relations. This led to improved model performance for 3D object mapping:

Figure 5: Comparison of visiPAM with human behavior. Violin plot comparing human placements and visiPAM predictions. Strip plots (small colored dots) indicate average distance of marker locations from the overall human mean, one point for each participant. Thin gray lines show the within-participant differences for the same- vs. different-superordinate-category conditions. Violin plots exclude data points greater than 2.5 standard deviations away from mean, though these individual data points are included in strip plots. Black horizontal lines indicate mean human distances in each condition, and error bars indicate standard deviations. Large green dots indicate visiPAM predictions (i.e., the distance between model predicted location and human mean location). Both human and visiPAM mappings were more variable when mapping objects from different superordinate categories.

But we agree that there is certainly still room for improvement. In our analyses of visiPAM’s errors, we have tried to indicate which directions are most promising for further improving visiPAM’s correspondence to human behavior. In particular, we believe that it will be beneficial to incorporate additional sources of relational information, including semantic, functional, and topological information. We have added discussion of these issues to the Discussion:

One limitation of the present work is the lack of a method for deriving 3D representations (such as point-clouds) from the 2D inputs that human reasoners typically receive. We found the incorporation of topological information (e.g., whether two parts are connected) significantly improved visiPAM’s mapping performance, but it would be desirable to derive this information directly from 2D inputs. Some recent advances may make it possible to take a step in that direction [39], but visiPAM will likely benefit from the rapid pace of innovation in representation learning algorithms. We are particularly excited about the continued development of general-purpose, self-supervised algorithms (such as the iBOT algorithm that we employed [21]), which we hope will provide increasingly rich representations over which reasoning

algorithms such as PAM can operate. Similarly, it will be useful in future work to incorporate non-visual semantic and functional information into visiPAM’s representations. We found that this kind of information was a significant contributing factor to human mapping judgments. Potential sources of such knowledge include word embeddings [40], multimodal embeddings [41], and representations of relations between verbal concepts [26].

[21] J. Zhou et al., “Ibot: Image bert pre-training with online tokenizer,” in 10th International Conference on Learning Representations, ICLR, 2022.

[26] H. Lu, Y. N. Wu, and K. J. Holyoak, “Emergence of analogy from relation learning,” Proceedings of the National Academy of Sciences, vol. 116, no. 10, pp. 4176–4181, 2019.

[39] R. Ranftl, A. Bochkovskiy, and V. Koltun, “Vision transformers for dense prediction,” in Proceedings of the IEEE/CVF International Conference on Computer Vision, 2021, pp. 12 179–12 188.

[40] T. Mikolov, I. Sutskever, K. Chen, G. S. Corrado, and J. Dean, “Distributed representations of words and phrases and their compositionality,” Advances in neural information processing systems, vol. 26, 2013.

[41] A. Radford et al., “Learning transferable visual models from natural language supervision,” in International conference on machine learning, PMLR, 2021, pp. 8748–8763.

Reviewer 3:

For 2D images, edges were determined by distance measures in the 2D plane. This seems like it could lead to large errors depending on how objects were positioned in 3D space? It was unclear to me what role the two distance measures played. Could the authors provide some intuition or even numerical results that informed this choice?

We have added ablation experiments demonstrating the importance of both types of edge information:

S3 Ablation analyses

S3.1 2D image mapping

VisiPAM’s edge embeddings involved two different types of spatial relations, one based on angular distance ($r\theta$), and one based on vector difference ($r\vec{d}$; see Section 4.2.3 for more details). We performed ablation experiments in which visiPAM’s edge embeddings only contained one of these spatial relation types. The results demonstrated that both components contributed significantly to visiPAM’s performance (Table S2).

	Within-category		Between-category
	Animals	Vehicles	Animals
VisiPAM	63.2% (59.1%)	69.5% (58.8%)	67.9% (59.9%)
VisiPAM (r_θ only)	56.7% (51.8%)	66.7% (55%)	67.5% (59.4%)
VisiPAM (r_δ only)	56.1% (51.2%)	65.4% (53.2%)	52.5% (40.6%)
Random	10%	26%	20%

Table S2: Analogical mapping with 2D images – edge embedding ablation analyses. Mapping accuracy on part-matching task. VisiPAM performed worse when edge embeddings were based only on angular distance (r_θ) or vector difference (r_δ). ‘Random’ denotes chance performance (determined by average number of part comparisons). Values in parentheses reflect chance-normalized performance (percentage of the range between chance performance and 100% accuracy).

However, we agree with the reviewer that visiPAM’s relational information could certainly be further improved. One particularly promising direction is to incorporate topological information, e.g. information about whether two object parts are connected. These kinds of relations may dissociate from spatial relations in some cases, especially for non-rigid-body objects such as animals. We have added the results of preliminary experiments demonstrating the usefulness of this kind of relational information:

We also carried out an additional experiment to determine whether visiPAM’s performance can be further improved by incorporating additional sources of relational information. Specifically, we augmented visiPAM’s edge embeddings with information about whether two parts are connected. This is an important aspect of relations between object parts that may not be entirely captured by simple spatial relations, especially for non-rigid body objects such as animals. For instance, the spatial relation between two body parts (e.g., the hand and the shoulder) can change dramatically depending on posture, but the topological relationship between body parts (e.g., which body parts are connected) remains unchanged. To test whether this information might further improve visiPAM’s performance, we tested a model in which edges were modified based on whether the corresponding object parts were connected (using groundtruth knowledge). For parts that were not connected, the values for the spatial relation components (r_θ and r_δ) were set to 0, while spatial relation embeddings were left unchanged for connected parts. Additionally, an extra dimension was added to the edge embeddings with a value of 1 for connected parts, and a value of -1 for non-connected parts. We found that this version of visiPAM performed even better than the version that included spatial relations only, particularly for mapping problems involving animals (Table S3). Though this preliminary experiment relied on groundtruth knowledge about the object parts, future work might explore methods for extracting this information directly from images.

Within-category		Between-category
Animals	Vehicles	Animals
72.4% (69.3%)	70.3% (59.9%)	77% (71.3%)

Table S3: VisiPAM’s performance is improved by incorporating topological relations. VisiPAM’s performance was improved when edges were augmented with information about whether two parts were connected. Values in parentheses reflect chance-normalized performance (percentage of the range between chance performance and 100% accuracy).

We also agree that it would be desirable to utilize 3D relations rather than 2D relations. This is currently very challenging given the difficulty of estimating depth position from monocular visual inputs (i.e., static, non-stereoscopic images). But we believe that advances in computer vision may soon make this possible. We have mentioned this possibility in the discussion:

One limitation of the present work is the lack of a method for deriving 3D representations (such as point-clouds) from the 2D inputs that human reasoners typically receive. We found the incorporation of topological information (e.g., whether two parts are connected) significantly improved visiPAM’s mapping performance, but it would be desirable to derive this information directly from 2D inputs. Some recent advances may make it possible to take a step in that direction [39], but visiPAM will likely benefit from the rapid pace of innovation in representation learning algorithms.

[39] R. Ranftl, A. Bochkovskiy, and V. Koltun, “Vision transformers for dense prediction,” in Proceedings of the IEEE/CVF International Conference on Computer Vision, 2021, pp. 12 179–12 188.

For the 2D case, why was the between category performance for the animals higher than within for visiPam (Table 2)? This result seems to conflict with the result shown in Figure 5.

For the 2D image task, there are fewer part comparisons in the between-category comparison (mapping from cat to horse) than in the within-category comparison (mapping from cat to cat, or horse to horse). This is because there are fewer common parts between different categories. Thus, performance in the between-category task needs to be assessed relative to a different baseline level for chance performance (chance performance is 20% accuracy in the between-category condition, and 10% accuracy in the within-category condition). We have clarified this issue in the revised paper, and have also added a measure that explicitly controls for this by normalizing performance based on the level of chance performance:

	Within-category		Between-category
	Animals	Vehicles	Animals
VisiPAM	63.2% (59.1%)	69.5% (58.8%)	67.9% (59.9%)
VisiPAM (nodes only)	55.5% (50.6%)	61.6% (48.1%)	62.3% (52.9%)
VisiPAM (edges only)	47.8% (42%)	63.7% (50.9%)	51.5% (39.4%)
SSMN	46.6% (40.7%)		
Random	10%	26%	20%

Table 1: Analogical mapping with 2D images. Mapping accuracy on part-matching task proposed by Choi et al. [23]. Original task in [23] involved evaluation on within-category animal comparisons after training with 37,330 mapping problems. VisiPAM significantly outperformed SSMN, the previous state-of-the-art, despite having no direct training on mapping. VisiPAM also performed well on new problems involving within-category vehicle comparisons, and between-category animal comparisons (e.g., mapping from cat to horse). VisiPAM performed best when mapping was based on both node and edge similarity. ‘Random’ denotes chance performance (determined by average number of part comparisons). **Values in parentheses reflect chance-normalized performance (percentage of the range between chance performance and 100% accuracy). VisiPAM’s performance is roughly comparable across all conditions once chance performance is taken into account.**

After controlling for this factor, performance on the 2D image mapping task is comparable in the within- vs. between-category conditions.

Additionally, in the 3D object mapping task, the conditions are defined based on whether the two objects are from the same *superordinate* category. For instance, the ‘same superordinate category’ condition involves comparisons between dogs and horses, which is similar to the ‘between-category’ condition in the 2D image mapping task. The ‘different superordinate category’ condition involves comparisons between very different kinds of objects, such as a dog and a chair. Thus, the ‘different superordinate category’ condition in the 3D object mapping task involves much more abstract comparisons than the ‘between-category’ condition in the 2D image mapping task.

It appears to be a limitation of the datasets used (e.g., Pascal Part Matching) that the number of classes compared is very small. Somewhat concerning, there appears to be differences between classes (e.g., Figure 5). On the positive side, there are a lot cases contained within these classes.

The difference between classes (increased variability for chair-to-chair mapping vs. animal-to-animal mapping) is most likely driven by the greater variety of chair shapes (as opposed to animal shapes) in our 3D object dataset. We have added a figure to the Supplementary Material illustrating this:

Figure S5: Explanation of increased variability in chair-to-chair condition. Both visiPAM and human participants displayed higher variability in the chair-to-chair condition than in the animal-to-animal condition. This was likely driven by the inherent diversity of chair shapes (see objects in Figure 6). To illustrate this, we have displayed two example problems. When the target and source chair had a similar shape (top panel), mappings had very low variability (heatmaps display distribution of human mapping judgments). When the target and source chair had very different shapes (bottom panel), mappings had much higher variability.

The complete set of objects are now displayed in Methods:

Figure 6: 3D Dataset. Eight chairs were selected from the ShapeNetPart dataset (each chair with a different shape), and eight animals from the Animal Pack dataset: horse, buffalo, Cane Corso, domestic pig, Celtic wolfhound, African elephant, Hellenic hound, and camel.

How does orientation, eccentricity, etc. affect performance? For example, if an image is reflected, is performance affected? Does the model suffer relative to humans when the cat in one image is facing the opposite direction as the cat in the other image? Is the model limited to

canonical orientations? More generally, the model works with images so I was expecting more consideration of how image properties affect performance with an eye toward identifying any shortcuts the model may be using. I.e., it looks like it's reasoning but is actually exploiting lower-level image information or confounds.

We have added an additional analysis testing how visiPAM performs when the target image is horizontally reflected. The results confirm that visiPAM's performance is unaffected by this factor. We have mentioned these results in section 2.2 of the revised paper:

Notably, we also found that visiPAM's performance was unaffected when the target image was horizontally reflected relative to the source (Supplementary Section S1.1 and Table S1), suggesting that the model is robust to low-level image manipulations.

And we have added these results to the Supplementary Material:

S1 Error analysis

S1.1 2d image mapping

We conducted additional analyses to better understand visiPAM's strengths and limitations in the context of the 2D image mapping task. First, we performed a test in which the target image was horizontally reflected relative to the source image. This experiment allowed us to test the extent to which visiPAM's performance might depend on canonical orientation. We found that this manipulation did not affect visiPAM's performance for either within-category or between-category animal mapping (Table S1), suggesting that the model is robust to these kinds of low-level image manipulations.

	Within-category	Between-category
	Animals	
VisiPAM (default)	63.2% (59.1%)	67.9% (59.9%)
VisiPAM (target horizontally reflected)	62.7% (58.6%)	67.9% (59.9%)
Random	10%	20%

Table S1: VisiPAM's performance is robust to image manipulations. Mapping accuracy for both within-category (e.g., mapping from cat to cat, or horse to horse) and between-category (e.g., mapping from cat to horse) animal comparisons was largely unaffected by horizontal reflection of the target image. 'Random' denotes chance performance (determined by average number of part comparisons in each condition). Values in parentheses reflect chance-normalized performance (percentage of the range between chance performance and 100% accuracy).

Is there good evidence for the model and people that mappings between images are systematic? For example, when the model gets a mapping correct, are the neighboring mappings more likely to be correct? This is related to the authors' discussion of bimodal solutions.

We have performed an analysis to address this question. The results indicate that both visiPAM and human participants are generally consistent in the mappings that they produce. These results have been added to the Supplementary Material:

Figure 3: Experiment measuring human performance for mapping between 3D objects. (a) Sample stimuli used in human experiment. Participants were instructed to move markers in target image to locations corresponding to same-color markers in source image. Left panel: example trials with source and target images from the same superordinate object category. Right panel: example trials with images from different superordinate categories. **(b)** Example heatmaps of human marker placements on target images for two comparisons. The intensity of the color indicates the proportion of participants who placed the marker in that location. Source images have been reduced in size for the purpose of illustration. Left panel: marker placements were highly consistent across subjects when source and target came from same superordinate category. Right panel: marker placement showed more variation across participants when source and target came from different superordinate categories.

S2 Consistency analysis

A noteworthy phenomenon is the consistent utilization of semantic, functional and visual information in human judgments when deciding the relative relation between the two marker locations. As illustrated in the right panel of Figure 3b (main text), this tendency resulted in the formation of two distinct clusters for each marker in the between-category condition, with one cluster reflecting semantic knowledge (mapping the red dot on the back of the chair to the back of the horse) and the other based on spatial similarity (mapping the back of the chair to the neck of the horse). Depending on the mapping chosen for the red dot (the back of the chair corresponds to the back or neck of the horse), the green dot was placed on different legs of the horse.

To systematically analyze whether humans make consistent judgments for the two marker placements, we conducted the following analysis. First, we employed a dip test (with a threshold of $p < 0.05$) to determine whether marker locations reported by human participants had a bimodal distribution. The dip test revealed 22 trials in which both red and green markers were bimodally distributed. For these trials, we used the KMeans++ algorithm [53] to segregate responses into two clusters. We proceeded to examine the consistency between the reported marker locations. In most of these 22 problems, we observed consistency in the type of information utilized by participants, with both markers being consistently matched based on either visual or semantic information.

Figure S6: Examples illustrating marker consistency. Two example problems are shown. For each problem, human mapping judgments were bimodal, with two clusters of responses for each of the green and red markers. These clusters were generally consistent with a strategy that was based on either spatial relational information or semantic information. The clusters labeled with the index 1 follow a semantic strategy, e.g. mapping the back of the dog to the back of the chair (left panel), or mapping the back of the chair to the back of the buffalo (right panel). The clusters labeled with the index 0 follow a spatial strategy, e.g. mapping the back of the dog to the seat of the chair. Human responses were most

often consistent, in the sense that both marker placements were governed either by a semantic strategy or a spatial strategy.

Figure S6 presents two examples that illustrate marker consistency. Each marker in the figure has two cluster centers with the same color, and a number plotted next to the cluster center indexing the cluster. Out of 41 participants, 15 placed the green and red markers in cluster 0 in the left panel of the four images, whereas 16 placed the markers in cluster 1; 10 other participants did not follow a semantic or visual pattern. In this context, cluster 0 represents mapping primarily based on geometric properties in the object, indicating that participants relied on visual similarity between the source and target objects. In contrast, cluster 1 is a mapping based on semantic knowledge: mapping the back of the dog to the back of the chair despite their different local geometry properties. When participants mapped the back of the dog to the back of the chair, the front of the chair is on the left of the chair image. Hence, the front leg of the dog (red dot) would map to the cluster-1 location of the chair.

Mapping consistency was also observed in the mapping between a chair and a buffalo (right panel). Specifically, out of the total of 41 participants, 13 performed mapping based on visual similarity, placing both the green and red markers in cluster 0. In contrast, 24 participants relied on semantic similarity, placing the markers in cluster 1. The other 4 participants did not show any apparent pattern in their judgments. These analyses revealed a general pattern of consistent mapping among the participants, with a majority utilizing semantic information for mapping. Overall, across the 22 between-category mapping problems (for which 41 participants yielded a total of 902 trials), 568 trials (63%) showed a consistent mapping strategy for human judgments.

An examination was also conducted to determine if visiPAM exhibited a consistent mapping strategy. The results revealed that in 19 out of the 22 problems, VisiPAM consistently applied the mapping strategy based on visual similarity. Thus, similar to human participants, visiPAM was generally consistent in the mappings that it produced.

One minor suggestion is to move Figure 1 into the Introduction. I didn't really get the basic idea of the model until digesting Figure 1.

We have implemented this change.

REVIEWERS' COMMENTS

Reviewer #1 (Remarks to the Author):

I reviewed the original submission and made several comments. I have carefully read the authors' responses and revisions, and I feel that they have addressed all of my concerns. The paper is now acceptable for publication.

Reviewer #2 (Remarks to the Author):

The author responses resolve most of my concerns of rich ablation study, detailed analysis and comparison to human performance. But about the novelty of the work, the authors' response does not resolve my concern of not sufficient novelty of the method rather to confirm that the method is a simple application of the PAM for vision task. Overall, the work is quite incremental in terms of technical contribution, it is a well designed method performing on par with humans.

Reviewer #3 (Remarks to the Author):

The authors consider mapping locations between visual images and 3D point clouds. For example, visiPam can map the head of a cat in one image to the head of a horse in another image. visiPam relies on a novel pairing of existing algorithms. The first stage is to extract node representations. Different approaches are used for 2D and 3D inputs. Edges then are defined by distance. PAM then does the node mapping based on a combination of satisfying node and edge similarity constraints. PAM is described as Bayesian, though it may be understood as performing graph matching by minimizing a cost function involving these two terms.

In my opinion, the authors did a good job in the revision and should be commended. The results are more completely reported and there are additional model evaluations, including ablation studies to better understand what is driving visiPAM's performance.

The demonstration in the reflecting simulations show that performance is not driven by images being in a canonical orientation, at least with respect to a horizontal flip.

Things that confused me, like differences between within vs. between category performance, were addressed (in this case in terms of making different guessing baselines clearer).

Of course, some inherent limitations in the work remain, such as the novelty of combining two existing models together and visiPAM performance levels. That said, I think the authors have made a reasonable effort in revision and to address these fundamental limitations would almost constitute a new contribution given the effort required. Therefore, in my opinion, a decision should be made on the manuscript. In the authors' favor, they address the fundamental limitations through reasonable discussion of visiPAM's limitations and how future efforts could help. In summary, the revision improves upon a good initial submission with no new weaknesses revealed.

We would like to thank the reviewers for their continued engagement and feedback on our manuscript. We are pleased that the reviewers found the revised manuscript significantly improved. We note that reviewers 2 and 3 expressed a remaining concern about the novelty of the proposed method. To address this issue, we clarified the extent to which our proposed model depends on previous work. Specifically we added the following additional text to the discussion section (new text in bold):

These results build on other recent work that has employed similarity-based reasoning over representations derived using deep learning [20, 26, 27], enabling analogy to emerge zero-shot. **Most notably, visiPAM builds on a previous model of analogical mapping, PAM [20], by combining it with embeddings from state-of-the-art computer vision models [21, 22], thus enabling its application to real-world visual inputs, even when those inputs are from distinctively different categories (i.e., far analogy).**

[20] H. Lu, N. Ichien, and K. J. Holyoak, "Probabilistic analogical mapping with semantic relation networks," *Psychological Review*, 2022.

[21] J. Zhou et al., "Ibot: Image bert pre-training with online tokenizer," in *10th International Conference on Learning Representations, ICLR, 2022*.

[22] Y. Wang, Y. Sun, Z. Liu, S. E. Sarma, M. M. Bronstein, and J. M. Solomon, "Dynamic graph cnn for learning on point clouds," *Acm Transactions On Graphics (tog)*, vol. 38, no. 5, pp. 1–12, 2019.

[26] H. Lu, Y. N. Wu, and K. J. Holyoak, "Emergence of analogy from relation learning," *Proceedings of the National Academy of Sciences*, vol. 116, no. 10, pp. 4176–4181, 2019.

[27] N. Ichien, Q. Liu, S. Fu, K. Holyoak, A. Yuille, and H. Lu, "Visual analogy: Deep learning versus compositional models," in *Proceedings of the 43rd Annual Meeting of the Cognitive Science Society, 2021*.